# LUZP1, a novel regulator of primary cilia and the actin cytoskeleton, is a contributing factor in Townes-Brocks Syndrome

Laura Bozal-Basterra[1], María Gonzalez-Santamarta[1†], Veronica Muratore[1], Aitor Bermejo-Arteagabeitia[1], Carolina Da Fonseca[1], Orhi Barroso-Gomila[1], Mikel Azkargorta[1,2,3], Ibon Iloro[1,2,3], Olatz Pampliega[4], Ricardo Andrade[5], Natalia Martín-Martín[1], Tess C Branon[6,7], Alice Y Ting[6,7,8], Jose A Rodríguez[9], Arkaitz Carracedo[1,10,11,12], Felix Elortza[1,2,3], James D Sutherland[1]*, Rosa Barrio[1]*

[1]Center for Cooperative Research in Biosciences (CIC bioGUNE), Basque Research and Technology Alliance (BRTA), Bizkaia Technology Park, Derio, Spain; [2]CIBERehd, Instituto de Salud Carlos III, Madrid, Spain; [3]ProteoRed-ISCIII, Instituto de Salud Carlos III, Madrid, Spain; [4]Department of Neurosciences, University of the Basque Country, Achucarro Basque Center for Neuroscience-UPV/EHU, Leioa, Spain; [5]Analytical & High Resolution Biomedical Microscopy Core Facility, University of the Basque Country (UPV/EHU), Leioa, Spain; [6]Department of Chemistry, Massachusetts Institute of Technology, Cambridge, United States; [7]Departments of Genetics, Chemistry and Biology, Stanford University, Stanford, United States; [8]Chan Zuckerberg Biohub, San Francisco, United States; [9]Department of Genetics, Physical Anthropology and Animal Physiology, University of the Basque Country (UPV/EHU), Leioa, Spain; [10]CIBERONC, Instituto de Salud Carlos III, Madrid, Spain; [11]Ikerbasque, Basque Foundation for Science, Bilbao, Spain; [12]Biochemistry and Molecular Biology Department, University of the Basque Country (UPV/EHU), Bilbao, Spain

*For correspondence:
jsutherland@cicbiogune.es (JDS);
rbarrio@cicbiogune.es (RB)

Present address: †ITAV-CNRS, Centre Pierre Potier Oncopole, Toulouse, France

Competing interests: The authors declare that no competing interests exist.

**Abstract** Primary cilia are sensory organelles crucial for cell signaling during development and organ homeostasis. Cilia arise from centrosomes and their formation and function is governed by numerous factors. Through our studies on Townes-Brocks Syndrome (TBS), a rare disease linked to abnormal cilia formation in human fibroblasts, we uncovered the leucine-zipper protein LUZP1 as an interactor of truncated SALL1, a dominantly-acting protein causing the disease. Using TurboID proximity labeling and pulldowns, we show that LUZP1 associates with factors linked to centrosome and actin filaments. Here, we show that LUZP1 is a cilia regulator. It localizes around the centrioles and to actin cytoskeleton. Loss of LUZP1 reduces F-actin levels, facilitates ciliogenesis and alters Sonic Hedgehog signaling, pointing to a key role in cytoskeleton-cilia interdependency. Truncated SALL1 increases the ubiquitin proteasome-mediated degradation of LUZP1. Together with other factors, alterations in LUZP1 may be contributing to TBS etiology.

## Introduction

Townes-Brocks Syndrome (TBS1 [MIM: 107480]) is an autosomal dominant genetic disease, caused by mutations in a transcription factor called SALL1, characterized by the presence of imperforate anus, dysplastic ears, thumb malformations and often renal and heart impairment, among other

**eLife digest** Primary cilia are the 'antennae' of animal cells: these small, flexible protrusions emerge from the surface of cells, where they help to sense and relay external signals. Cilia are assembled with the help of the cytoskeleton, a dynamic network of mesh-like filaments that spans the interior of the cell and controls many different biological processes. If cilia do not work properly, human diseases called ciliopathies can emerge.

Townes-Brocks Syndrome (TBS) is an incurable disease that presents a range of symptoms such as malformations of the toes or fingers, hearing impairment, and kidney or heart problems. It is caused by a change in the gene that codes for a protein called SALL1, and recent work has also showed that the cells of TBS patients have defective cilia. In addition, this prior research identified a second protein that interacted with the mutant version of SALL1; called LUZP1, this protein is already known to help maintain the cytoskeleton.

In this study, Bozal-Basterra et al. wanted to find out if LUZP1 caused the cilia defects in TBS. First, the protein was removed from mouse cells grown in the laboratory, which dramatically weakened the cytoskeleton. In keeping with this observation, both the number of cilia per cell and the length of the cilia were abnormal. Cells lacking LUZP1 also had defects in a signalling process that transmits signals received by cilia to different parts of the cell. All these defects were previously observed in cells isolated from TBS patients. In addition, LUZP1-deficient mouse cells showed the same problems with their cilia and cytoskeleton as the cells from individuals with TBS. Crucially, the cells from human TBS patients also had much lower levels of LUZP1 than normal, suggesting that the protein may contribute to the cilia defects present in this disease.

The work by Bozal-Basterra et al. sheds light on how primary cilia depend on the cytoskeleton, while also providing new insight into TBS. In the future, this knowledge could help researchers to develop therapies for this rare and currently untreatable disease.

symptoms (*Botzenhart et al., 2007*; *Kohlhase et al., 1998*). Some of these features overlap those in the ciliopathic spectrum. It has been recently demonstrated that primary cilia defects are contributing factors to TBS etiology (*Bozal-Basterra et al., 2018*). Truncated SALL1, either by itself or in complex with full length protein (SALL1$^{FL}$), can interact with CCP110 and CEP97. As a consequence, those negative regulators are reduced at the mother centriole (MC) and ciliogenesis is promoted (*Bozal-Basterra et al., 2018*).

Primary cilia are sensory organelles that have a crucial role in cell signaling and protein trafficking during development and organ homeostasis. Although several key pathways are influenced by cilia function (Wnt, TGFbeta, PDGFRalpha, Notch), the best characterized is the Sonic Hedgehog (Shh) pathway (*Goetz and Anderson, 2010*). Briefly, Shh binds to its receptor PTCH1 and leads to ciliary enrichment of the transmembrane protein Smoothened (SMO), with concomitant conversion of the transcription factor GLI3 from a cleaved repressor to a full-length activator form, leading to activation of Shh target genes. Two such genes are *PTCH1* and *GLI1* (encoding the Shh receptor and a transcriptional activator, respectively), exemplifying the feedback and fine-tuning of the Shh pathway.

Cilia arise from the centrosome, a cellular organelle composed of two barrel-shaped microtubule-based structures called the centrioles. Primary cilia formation is very dynamic throughout the cell cycle. Cilia are nucleated from the MC at the membrane-anchored basal body upon entry into the G0 phase, and they reabsorb as cells progress from G1 to S phase, completely disassembling in mitosis (*Rezabkova et al., 2016*). Centrioles are surrounded by protein-based matrix, the pericentriolar material (PCM) (*Conduit et al., 2015*; *Vertii et al., 2016*). In eukaryotic cells, PCM proteins are concentrically arranged around a centriole in a highly organized manner (*Fu and Glover, 2012*; *Lawo et al., 2012*; *Mennella et al., 2012*; *Sonnen et al., 2012*). Based on this observation, proper positioning and organization of PCM proteins may be important for promoting different cellular processes in a spatially regulated way (*Kim et al., 2019*). Not surprisingly, aberrations in the function of PCM scaffolds are associated with several human diseases, including cancer and ciliopathies (*Gönczy, 2015*; *Nigg and Holland, 2018*). Cilia assembly is regulated by diverse factors. Among them, CCP110 and CEP97 form a cilia suppressor complex that, when removed from the MC, allows

ciliogenesis to proceed (*Spektor et al., 2007*). The actin cytoskeleton is also emerging as key regulator of cilia formation and function, with both negative and positive roles (*Copeland, 2020*).

Ciliary dysfunction often results in early developmental problems including hydrocephalus, neural tube closure defects (NTD) and left-right anomalies (*Fliegauf et al., 2007*). These features are often reported in a variety of diseases, collectively known as ciliopathies, caused by failure of cilia formation and/or cilia-dependent signaling (*Hildebrandt et al., 2011*). In the adult, depending on the underlying mutation, ciliopathies present a broad spectrum of phenotypes comprising cystic kidneys, polydactyly, obesity or heart malformation.

Truncated SALL1 likely interferes with multiple factors to give rise to TBS phenotypes. Here we focus on LUZP1, a leucine-zipper motif containing protein that was identified by proximity proteomics as an interactor of truncated SALL1 (*Bozal-Basterra et al., 2018*). LUZP1 has been previously identified as an interactor of ACTR2 (ARP2 actin related protein two homologue) and filamin A (FLNA) and, recently, as an actin cross-linking protein (*Hein et al., 2015*; *Wang and Nakamura, 2019*). Furthermore, LUZP1 shows homology to FILIP1, a protein interactor of FLNA and actin (*Gad et al., 2012*; *Nagano et al., 2004*). Interestingly, mutations in *Luzp1* resulted in cardiovascular defects and cranial NTD in mice (*Hsu et al., 2008*), phenotypes within the spectrum of those seen in TBS individuals and mouse models of dysfunctional cilia (*Botzenhart et al., 2007*; *Botzenhart et al., 2005*; *Klena et al., 2016*; *Kohlhase et al., 1998*; *Surka et al., 2001*; *Toomer et al., 2019*). Both the non-canonical Wnt/PCP (Wingless-Integrated/planar cell polarity) and the Shh pathways are influenced by the presence of functional cilia and regulate neural tube closure and patterning (*Campbell, 2003*; *Copp, 2005*; *Fuccillo et al., 2006*). Remarkably, ectopic Shh was observed in the dorsal lateral neuroepithelium of the *Luzp1⁻/⁻* mice (*Hsu et al., 2008*). However, in spite of the phenotypic overlaps, a link between LUZP1 and ciliogenesis has not been explored.

Here we demonstrate that LUZP1 is associated with centrosomal and actin cytoskeleton-related proteins. We show that LUZP1 localizes to the PCM, actin cytoskeleton and the midbody, and also provide evidence towards its regulatory role on actin dynamics and its subsequent impact on ciliogenesis. Notably, we demonstrate that *Luzp1⁻/⁻* cells exhibit reduced filamentous actin (F-actin), longer primary cilia, higher rates of ciliogenesis and increased Shh signaling. Furthermore, TBS-derived primary fibroblasts show a reduction in LUZP1 and actin filaments, possibly through SALL1-regulated LUZP1 degradation via the ubiquitin (Ub)-proteasome system (UPS). As a novel regulator of ciliogenesis and the actin cytoskeleton, LUZP1 might contribute to the aberrant cilia phenotype in TBS.

## Results

### SALL1 interacts with LUZP1

Using proximity proteomics, we have previously shown that a truncated and mislocalized form of SALL1 present in TBS individuals (SALL1²⁷⁵) can interact aberrantly with cytoplasmic proteins (*Bozal-Basterra et al., 2018*). LUZP1 was found among the most enriched proteins in the SALL1²⁷⁵ proximal interactome. We confirmed this finding by independent BioID experiments analyzed by western blot using a LUZP1-specific antibody (*Figure 1A* and *Figure 1—figure supplement 1*). To further characterize the interaction of LUZP1 with truncated SALL1, we performed pulldowns using tagged SALL1²⁷⁵-YFP in HEK 293FT cells. Our results showed that endogenous LUZP1 was able to interact with SALL1²⁷⁵, confirming our proximity proteomics data (*Figure 1—figure supplement 2A*, lane 6, and *Figure 1—figure supplement 1*). The interaction with SALL1²⁷⁵ persisted in presence of overexpressed *SALL1^FL* (*Figure 1—figure supplement 2A*, lane 9, and *Figure 1—figure supplement 1*), suggesting that heterodimerization of the truncated and FL forms does not inhibit the interaction with LUZP1. When expressed alone, we noted that SALL1^FL-YFP also interacts with LUZP1 in pulldown assays (*Figure 1—figure supplement 2A* and *Figure 1—figure supplement 1*). As these proteins have distinct localizations (nuclear and cytoplasmic, respectively), the interaction likely occurs in post-lysis cell extracts (more in Discussion). These results show that the truncated form of SALL1 expressed in TBS individuals, either by itself or in complex with the FL form, can interact with LUZP1.

### LUZP1 interacts with centrosomal and actin cytoskeleton components

To gain insights into the function of LUZP1, we sought to identify its proximal interactome using the TurboID approach (*Branon et al., 2018*). We used RPE1 cells stably expressing low levels of FLAG-

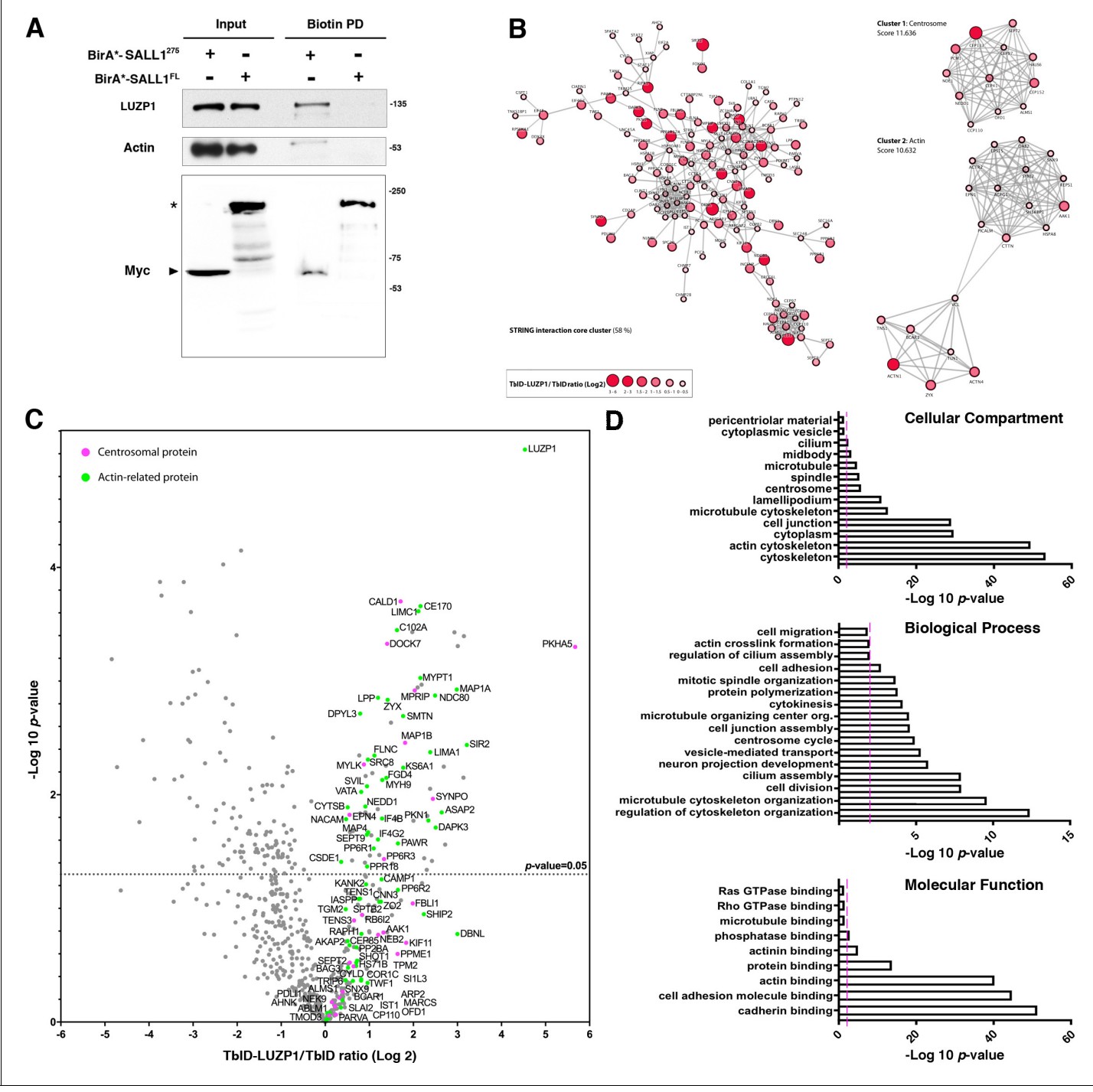

**Figure 1.** Proximity proteomics reveal LUZP1 interaction with truncated SALL1 and with centrosome- and actin cytoskeleton-associated proteins. (**A**) Western blot analysis of BioID, streptavidin pulldown (PD) of HEK 293FT cells transfected with Myc-tagged BirA*-SALL1[275] or BirA*-SALL1[FL]. Specific antibodies (LUZP1, actin, Myc) were used as indicated. Anti-Myc antibody detected the self-biotinylated form of BirA*-SALL1[FL] (asterisk) or BirA*-SALL1[275] (black arrowhead). (**B**) STRING core cluster network analysis of LUZP1 interactors (confidence 0.7 or higher), visualized using Cytoscape software. Color and size of the nodes indicate Log2 of the TurboID-LUZP1 (TbID-LUZP1) *versus* TbID alone ratio. The most highly interconnected clusters, centrosome and actin, are indicated separately. (**C**) Volcano plot representing the distribution of the candidates identified by proximity proteomics in three independent experiments. Proteins with more than 1-fold change in TbID-LUZP1 intensity with respect to the TbID (Log$_2 \geq 0$) were considered as LUZP1-associated candidates (grey dots). Proteins associated with the actin cytoskeleton and the centrosome were colored in green and pink colors, respectively. (**D**) Graphical representation of the -Log$_{10}$ of the *p*-value for each of the represented GO terms of the TbID experiment performed on RPE1 stably expressing near endogenous levels of TbID-LUZP1 *vs* TbID. Pink dotted line represents the cutoff of p-value<0.01.

*Figure 1 continued on next page*

*Figure 1 continued*

The online version of this article includes the following figure supplement(s) for figure 1:

**Figure supplement 1.** Full western blot images for *Figure 1*.
**Figure supplement 2.** Analysis of TbID constructs expression and biotinylation localization.

TurboID-LUZP1 (TbID-LUZP1) or FLAG-TurboID (TbID) as control. Transduced cells showed sub-endogenous expression levels of TbID-LUZP1 (*Figure 1—figure supplement 2B*, two asterisks; endogenous, two open arrowheads). Staining of transfected cells revealed that, proteins biotinylated by TbID are diffusely localized throughout the nucleus and cytoplasm, whereas those biotinylated by TbID-LUZP1 are localized primarily at the centrosome and actin cytoskeleton, as shown by fluorescent streptavidin (*Figure 1—figure supplement 2C*). Total lysates from TbID-LUZP1 or TbID-expressing cells were subjected to streptavidin pulldown and isolated proteins were analyzed by liquid chromatography tandem mass spectrometry (LC-MS/MS). 234 high-confidence proximity LUZP1 interactors were enriched in the TbID-LUZP1 *vs* the TbID proteome in three replicates (Source Data 1). Proteins enriched among the identified LUZP1 proximal interactors were centrosomal and actin cytoskeleton-related proteins (*Figure 1B–C*). With the purpose of obtaining a functional overview of the main pathways associated to LUZP1, a comparative Gene Ontology (GO) analysis was performed with all the hits enriched in TbID-LUZP1 *versus* TbID cells. In the Cellular Component domain, 'actin cytoskeleton', 'microtubule cytoskeleton', 'centrosome', 'cilium' and 'midbody' terms were highlighted among others (*Figure 1D* and Source Data 1). In the category of Biological Process, LUZP1 proteome showed enrichment in the 'microtubule cytoskeleton organization', 'cell division', 'cilium assembly', 'vesicle-mediated transport', 'microtubule organizing center organization' and 'cell adhesion' categories among others (*Figure 1D* and Source Data 1). With respect to Molecular Function, LUZP1 also showed enrichment in cytoskeleton-related proteins ('actin binding', 'actinin binding' and 'microtubule binding' terms; *Figure 1D* and Source Data 1). 37 or 96 of the verified or potential, respectively, centrosome/cilia gene products previously identified by proteomic studies (*Alves-Cruzeiro et al., 2014*; *Gupta et al., 2015*) were found as LUZP1 proximal interactors, supporting the enrichment of centrosome-related proteins among the potential interactors of LUZP1. In addition, 45 of LUZP1 proximal interactors were present among the actin-localized proteins identified by the Human Protein Atlas project based on subcellular localization to actin filaments (*Uhlen et al., 2015*).

## LUZP1 localizes to the centrosome, actin cytoskeleton and midbody

We examined the subcellular localization of LUZP1 in diverse cell types. First, immunostainings showed that endogenous LUZP1 surrounds both centrioles, labelled by centrin-2 (CETN2) in human RPE1 cells (*Figure 2A*) and gamma-tubulin in human dermal fibroblasts (*Figure 2—video 1*) (for specificity of LUZP1 antibody, check Figure 5). We examined LUZP1 localization at the centrosome in synchronized RPE1 cells. LUZP1 was reduced at the centrosome during G2/M and G0 phases (*Figure 2—figure supplement 1*). Interestingly, LUZP1 levels increased upon treatment with the proteasome inhibitor MG132 in G0 phase arrested-RPE1 cells. All together, these results indicate that LUZP1 levels are reduced at the centrosome in G2/M phase and upon starvation, and this reduction might be mediated by the UPS. The localization of LUZP1 at the centrosome was reproduced in U2OS cells expressing *LUZP1-YFP* (*Figure 2B–D*). We did not observe colocalization of LUZP1 with the distal centriolar marker CCP110 (*Figure 2B*), indicating that LUZP1 is likely found at the proximal end of both centrioles. We further imaged LUZP1 along with pericentrin (PCNT) and PCM1, markers of PCM. Interestingly, we observed that LUZP1 enveloped PCNT (*Figure 2C*), with LUZP1 itself being surrounded by PCM1 (*Figure 2D*). These results suggest that LUZP1 might be a novel PCM associated-protein, decorating the proximal end of both centrioles. In concordance with this localization, LUZP1 was associated to PCM1 in TurboID experiments (Source Data 1).

In addition to the localization at the centrosome/basal body, LUZP1 also localized to actin stress fibers (*Figure 2E*) and to the midbody (*Figure 2F*) in U2OS cells.

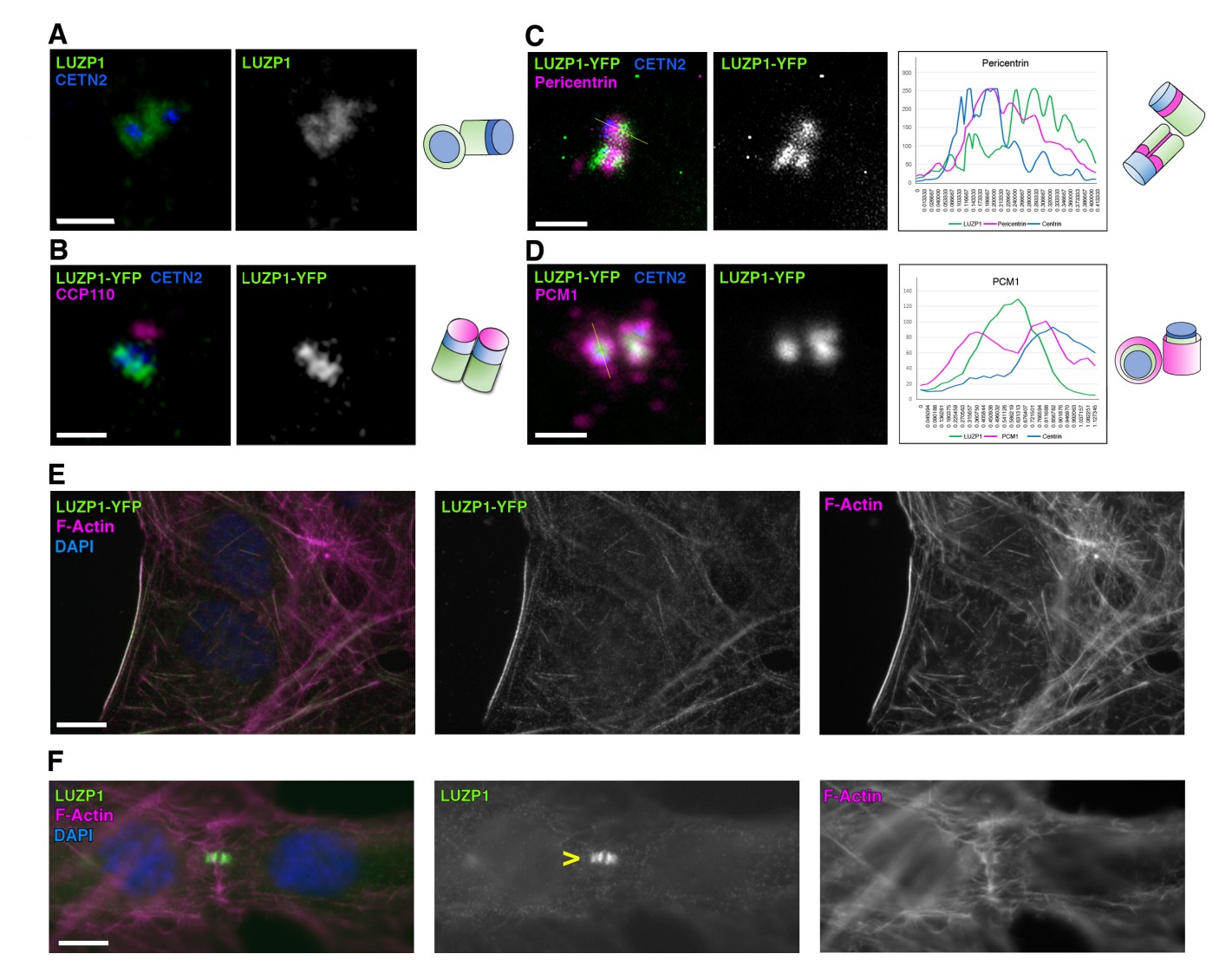

**Figure 2.** LUZP1 localizes to the centrosome and the actin cytoskeleton. (A) Immunofluorescence micrographs of LUZP1 (green) and Centrin-2 (CETN2, blue) in RPE1 cells. (B–D) Immunofluorescence micrographs of U2OS cells expressing *LUZP1-YFP* (green) stained with antibodies against CETN2 (blue) and CCP110 (B), Pericentrin (C) and PCM1 (D) in magenta. Plot profile of the different fluorophore intensities along the yellow lines in (C, D). Schematic representation of LUZP1 localization at the centrosome according to their respective micrographs in (A–D). Scale bar, 1 μm. Imaging in (A–D) was performed using confocal microscopy (Leica SP8, 63x objective). Lightning software (Leica) was applied. (E, F) Immunofluorescence micrographs of U2OS cells stained with an antibody against endogenous LUZP1 (green), phalloidin to detect F-actin (magenta), and counterstained with DAPI (blue). Single green and magenta channels are shown in black and white. (F) LUZP1 at the midbody in dividing cells (yellow arrowhead). Scale bar, 10 μm (E, F). Imaging in (E, F) was performed using widefield fluorescence microscopy (Zeiss Axioimager D1, 63x objective).

The online version of this article includes the following video and figure supplement(s) for figure 2:

**Figure supplement 1.** Centrosomal localization of LUZP1 at different cell cycle stages.

**Figure 2—video 1.** LUZP1 localization in the centrosome.

https://elifesciences.org/articles/55957#fig2video1

## LUZP1 shows reduction at the centrosome in TBS fibroblasts and interacts with centrosome-associated proteins

Based on the LUZP1 interaction with truncated SALL1, we checked its subcellular localization in fibroblasts derived from a TBS individual (TBS[275]; see Materials and methods) as well as non-TBS control. We observed that LUZP1 was markedly decreased at the centrosome of TBS[275] cells

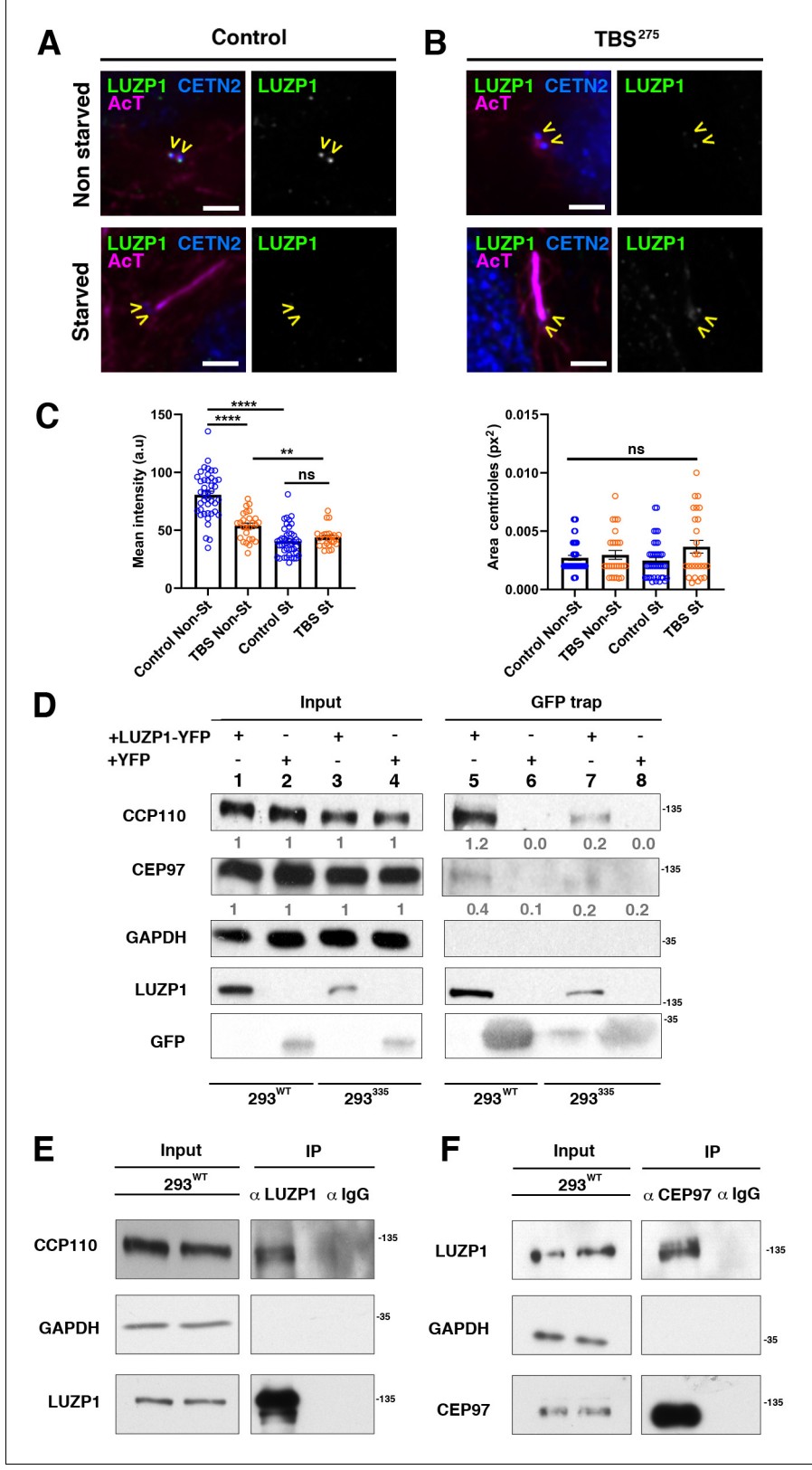

**Figure 3.** TBS cells show reduced LUZP1 levels at the centrosome. (**A, B**) Immunofluorescence micrographs of non-starved and starved human-derived Control (**A**) and TBS[275] fibroblasts (**B**) stained with antibodies against endogenous LUZP1 (green, yellow arrowheads), Centrin-2 (CETN2, blue) and acetylated alpha-tubulin (magenta). Black and white images show the isolated green channel. Note the reduction of LUZP1 in starved cells and in non-

*Figure 3 continued on next page*

*Figure 3 continued*

starved TBS[275] compared to non-starved control fibroblasts. Imaging was performed using widefield fluorescence microscopy (Zeiss Axioimager D1, 63x objective). Scale bar, 4 μm. (C) Graphical representation of the LUZP1 mean intensities (left panel) or the centrosome area (right panel), corresponding to the experiments shown in (A–B); n ≥ 6 micrographs. Three independent experiments were pooled together. *P*-values were calculated using the unpaired two-tailed Student´s test or U- Mann-Whitney test. (D) Western blot of inputs (lanes 1 to 4) and GFP-Trap pulldowns (lines 5 to 8) performed in WT HEK 293FT cells or in 293[335]*SALL1* mutant cells transfected with *LUZP1-YFP* (lanes 1, 3, 5 and 7) or *YFP* alone (lanes 2, 4, 6 and 8). Numbers under CCP110 and CEP97 panels result from dividing band intensities of each pulldown by their respective input levels. GAPDH was used as loading control. (E, F) Co-immunoprecipitation experiments show LUZP1-CCP110 (E) and CEP97-LUZP1 (F) interactions. Rabbit IgG was used for immunoprecipitation controls. GAPDH was used as loading and specificity control. In all panels, specific antibodies (LUZP1, GAPDH, CCP110, CEP97, GFP) were used as indicated. Blots shown here are representative of three independent experiments. Molecular weight markers are shown to the right.

The online version of this article includes the following figure supplement(s) for figure 3:

**Figure supplement 1.** Full western blot images for *Figure 3*.

---

compared to control cells in non-starved conditions (*Figure 3A–C*). Furthermore, LUZP1 levels decreased in starved *vs* non-starved control cells (*Figure 3A–C*), while centrosomal size remained unaltered (*Figure 3C*, right panel). We previously found that SALL1[275]-YFP interacted with the centrosome-associated ciliogenesis suppressors, CCP110 and CEP97 (*Bozal-Basterra et al., 2018*), so we checked whether LUZP1 could also interact with these factors. Indeed, LUZP1-YFP interacts with CCP110 and CEP97 in both WT (293[WT]) and TBS model (293[335]) HEK 293FT cells (*Figure 3D*, lanes 5 and 7, respectively and *Figure 3—figure supplement 1*; *Bozal-Basterra et al., 2018*). Less CCP110 and CEP97 was recovered in LUZP1-YFP pulldowns from 293[335] cells, but this is likely due to the reduced LUZP1-YFP seen in those cells (*Figure 3D*, Input, lanes 1 and 2 *vs* lane 3 and 4 and *Figure 3—figure supplement 1*). Beyond pulldowns, we found that immunoprecipitation of endogenous LUZP1 led to co-purification of endogenous CCP110 (*Figure 3E* and *Figure 3—figure supplement 1*) and that anti-CEP97 antibodies immunoprecipitated endogenous LUZP1 (*Figure 3F* and *Figure 3—figure supplement 1*). Both CCP110 and CEP97 were also identified as proximal interactors of TbID-LUZP1 (Source Data 1). Immunofluorescent colocalization on centrioles of LUZP1 (proximal) and CCP110 (distal) was not evident, suggesting that the interaction is indirect or occurs before proteins reach their destinations. However, these key regulators of ciliogenesis are just two of multiple centrosomal proteins associated with LUZP1, suggesting that LUZP1 may have a function at this dynamic organelle.

## LUZP1 localizes to actin and is altered in TBS fibroblasts

In addition to the centrosome/basal body, LUZP1 also localized to actin stress fibers, as well as the midbody in dividing cells (*Figure 2*). Intriguingly, when LUZP1 levels were examined in TBS[275] cells, a reduction in both actin-associated LUZP1 and phalloidin-labelled stress fibers was observed when compared to control cells (*Figure 4A–C*). These results indicate that actin cytoskeleton might be altered in TBS cells. Using pulldown assays, we confirmed that LUZP1-YFP interacts with both actin and FLNA (*Figure 4D* and *Figure 4—figure supplement 1*). Notably, actin, FLNA, alpha-actinin, palladin, LIMA1/Eplin and other stress fiber-associated proteins are proximal interactors of TbID-LUZP1 (Source Data 1). To examine whether LUZP1 levels change upon F-actin perturbation, HEK 293FT cells were treated with Cytochalasin D (CytoD), an inhibitor of actin polymerization. No changes in LUZP1 levels upon actin depolymerization were observed when cells were lysed in strong lysis conditions (WB5) (*Figure 4E–F* and *Figure 4—figure supplement 1*). However, we observed increased LUZP1 levels using mild lysis conditions (0.1% Triton X-100; *Figure 4E–F* and *Figure 4—figure supplement 1*). These results reflect that the integrity of the actin cytoskeleton may influence the solubility but not the stability of LUZP1.

## LUZP1 plays a role in primary cilia formation and shh signaling

Based on the localization of LUZP1 at the centrosome, its interaction with centrosomal proteins and the defects in ciliogenesis previously observed in TBS cells (*Bozal-Basterra et al., 2018*), we hypothesized that LUZP1 might have a role in cilia formation. To examine this, we analyzed ciliogenesis in

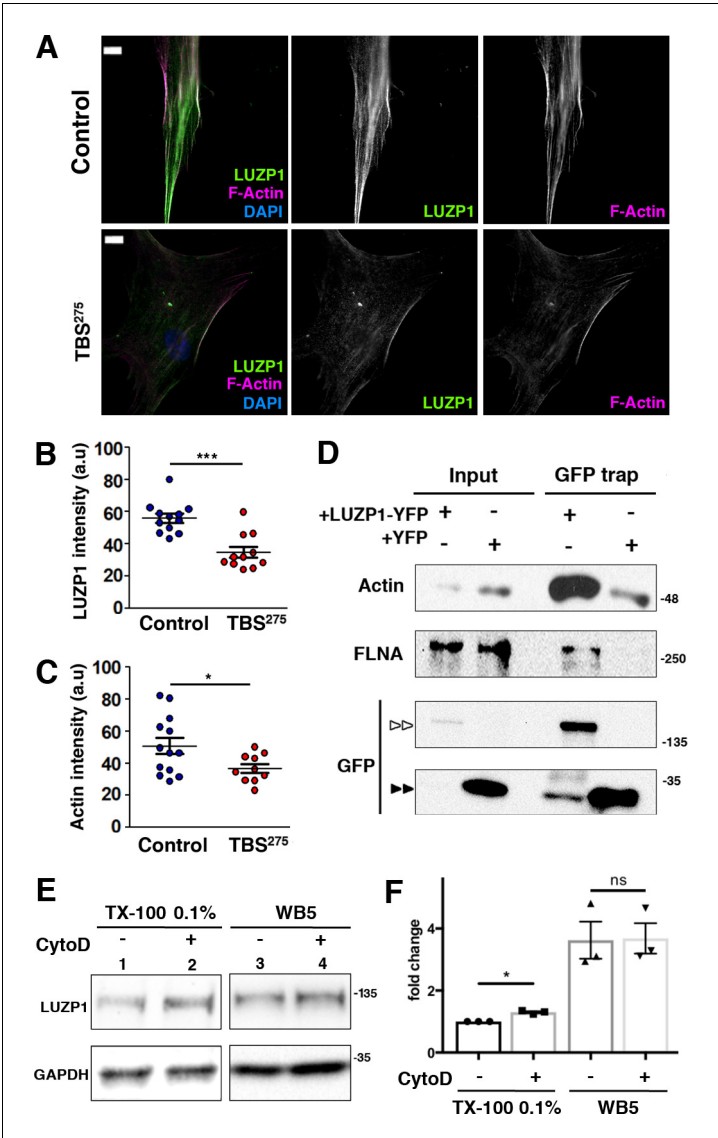

**Figure 4.** Reduction in LUZP1 coincides with decreased F-actin. (A) Immunofluorescence micrographs of Control and TBS[275] human fibroblasts stained with an antibody against endogenous LUZP1 (green), phalloidin to label F-actin (magenta), and counterstained with DAPI to label the nuclei (blue). Black and white images show the single green and magenta channels. Note the overall reduction in LUZP1 and F-actin levels in TBS[275] compared to control fibroblasts. Scale bar, 10 µm. Imaging was performed using widefield fluorescence microscopy (Zeiss Axioimager D1, 63x objective). (B, C) Graphical representation of the LUZP1 (B) and F-actin (C) mean intensities, corresponding to the experiments shown in (A); n ≥ 6 micrographs. Three independent experiments were pooled together. *P*-values were calculated using the unpaired two-tailed Student´s test or U- Mann-Whitney test. (D) Western blot of inputs or GFP-Trap pulldowns performed in HEK 293FT cells transfected with *LUZP1-YFP* or *YFP* alone. Anti-GFP antibody detected YFP alone (two black arrowheads) and LUZP1-YFP (two white arrowheads). Blots shown here are representative of three independent experiments. Molecular weight markers are shown to the right. Specific antibodies (LUZP1, GAPDH, CCP110, CEP97, GFP) were used as indicated. (E) Western blot of total cell lysates of HEK 293FT treated or not with cytochalasin D (CytoD) in a mild lysis buffer (TX-100 0.1%, lanes 1, 2) or a strong lysis buffer (WB5, lanes 3, 4). Note the increase in LUZP1 levels upon actin polymerization blockage with CytoD, exclusively when cells were lysed on 01% TX-100-based lysis buffer. GAPDH was used as loading control. In (D) and (E) panels, specific antibodies (LUZP1, GAPDH, actin, FLNA, GFP) were used as indicated. (F) Graphical representation of LUZP1 *vs* GAPDH band intensities in (E) normalized to lane 1. Graphs represent Mean and SEM of three independent experiments. *P*-value was calculated using two tailed unpaired Student´s t-test. Molecular weight markers in (D) and (E) are shown to the right.

The online version of this article includes the following figure supplement(s) for figure 4:

*Figure 4 continued on next page*

Shh-LIGHT2 cells, a cell line derived from immortalized mouse NIH3T3 fibroblasts that can display primary cilia and report on Shh pathway status using integrated luciferase reporters (herein designated as WT) (*Taipale et al., 2000*). Using CRISPR/Cas9 gene editing directed to exon 1 of murine *Luzp1*, we generated Shh-LIGHT2 mouse embryonic fibroblasts null for *Luzp1* (Luzp1$^{-/-}$ cells). For genetic rescue experiments, LUZP1 was restored to these cells by the expression of human *LUZP1-YFP* fusion (+LUZP1 cells). To examine the effect of the *Luzp1* mutation and rescue strategies, we used anti-LUZP1 antibody and checked LUZP1 localization associated with the actin cytoskeleton and the centrosome by immunofluorescence (*Figure 5A,B*), and its levels of expression by western blot (*Figure 5C* and *Figure 5—figure supplement 1*) in WT, Luzp1$^{-/-}$ and +LUZP1 cells. These experiments showed the effectiveness of the knockout and rescue strategies.

To analyze the role of LUZP1 in ciliation, WT, Luzp1$^{-/-}$ and +LUZP1 cells were plated at equal densities and induced to ciliate for 48 hr by serum withdrawal (starved; *Figure 6A*). We quantified ciliation rates and primary cilia length in non-starved and starved cells. Luzp1$^{-/-}$ fibroblasts displayed higher ciliation rate (60%) than WT (10.5%) and +LUZP1 (22.2%) when the cells were not subjected to starvation (*Figure 6B*). However, Luzp1$^{-/-}$ cells were not significantly more ciliated than WT or

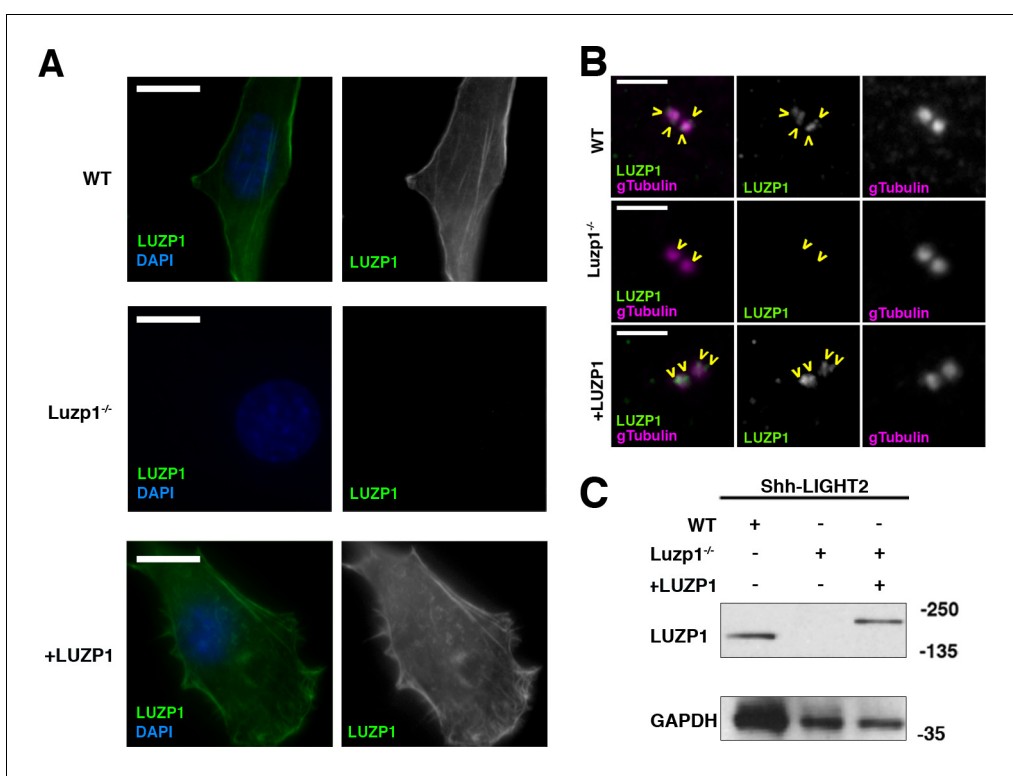

**Figure 5.** Generation of LUZP1 mutant cells. (**A, B**) Immunofluorescence micrographs of Shh-LIGHT2 control cells (WT), *Luzp1* depleted Shh-LIGHT2 cells (Luzp1$^{-/-}$) and Luzp1$^{-/-}$ cells rescued with human *LUZP1* (+LUZP1 cells) stained with a specific antibody against endogenous LUZP1 (green) and DAPI (blue) (**A**) or gamma-tubulin (magenta) (**B**). Single green and magenta channels are shown in black and white. Scale bars, 10 μm (**A**) or 2.5 μm (**B**). Images were taken using widefield fluorescence microscopy (Zeiss Axioimager D1, 63x objective). Note the lack of LUZP1 in Luzp1$^{-/-}$ cells. (**C**) Western blot analysis of total lysates of WT, Luzp1$^{-/-}$ and +LUZP1 cells using anti-LUZP1 antibodies. Molecular weight markers are shown to the right. Note the lack of LUZP1 signal in Luzp1$^{-/-}$ cells by immunofluorescence and western blot.

The online version of this article includes the following figure supplement(s) for figure 5:

**Figure supplement 1.** Full western blot images for *Figure 5*.

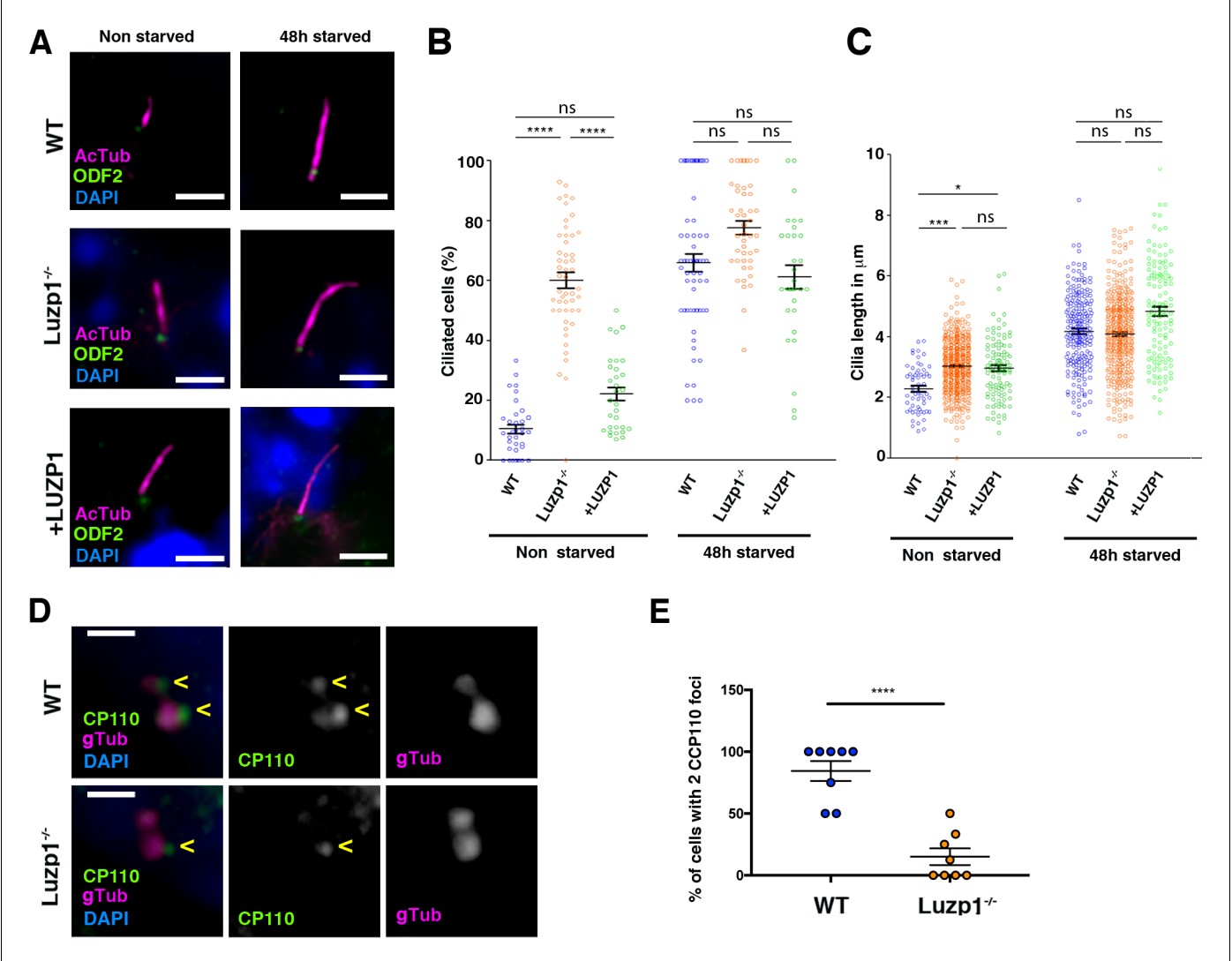

**Figure 6.** Loss of Luzp1 causes aberrant cilia frequency and length and Shh signaling. (**A**) Micrographs of Shh-LIGHT2 cells (WT), Shh-LIGHT2 cells lacking *Luzp1* (Luzp1[-/-]) and Luzp1[-/-] cells rescued with human *LUZP1-YFP* (+LUZP1) analyzed in cycling conditions (non-starved), or during cilia assembly (48 hr starved). Cilia were visualized by acetylated alpha-tubulin (magenta), basal body by ODF2 (green) and nuclei by DAPI (blue). Scale bar 2.5 μm. (**B, C**) Graphical representation of percentage of ciliated cells per micrograph (**B**) and cilia length (**C**) measured in WT (blue circles, n > 34 micrographs), Luzp1[-/-] (orange circles, n > 44 micrographs) or +LUZP1 cells (green circles, n > 30 micrographs) from three independent experiments. No starvation: WT 2.3 μm; Luzp1[-/-] cells 3.0 μm; +LUZP1 cells 2.9 μm; 48 hr starvation: WT 4.2 μm; Luzp1[-/-] cells 4.1 μm; +LUZP1 cells 4.8 μm; all average measures. (**D**) Immunofluorescence micrographs of WT and LUZP1[-/-] cells stained with antibodies against endogenous CCP110 (green), gamma-tubulin (gTub) to label the centrioles (magenta) and DAPI to label the nuclei (blue). Black and white images show the single green and magenta channels. Note the different distribution of CCP110 to the centrosome in LUZP1[-/-] compared to WT cells. Scale bar, 1 μm. (**E**) Graphical representation of the percentage of cells showing the presence of CCP110 to both centrioles per micrograph corresponding to the experiments in (**D**); n = 10 micrographs. Three independent experiments were pooled together.

+LUZP1 fibroblasts upon 48 hr of starvation (*Figure 6B*). In addition, primary cilia in Luzp1[-/-] cells were significantly longer than in non-starved WT cycling cells (*Figure 6A and C*), while under starvation the differences were not significant (*Figure 6A and C*). Note that, differently than the ciliation rate, cilia length was not rescued by adding human LUZP1. Taken together, these results confirm that Luzp1[-/-] cells display longer and more abundant primary cilia compared to WT cells in cycling conditions and indicate that LUZP1 might affect primary cilia dynamics.

One key event in ciliogenesis is the depletion of CCP110 and its partner CEP97 from the distal end of the MC, promoting the ciliary activating program in somatic cells (*Goetz et al., 2012*;

*Kleylein-Sohn et al., 2007*; *Prosser and Morrison, 2015*; *Spektor et al., 2007*; *Tsang et al., 2008*). We analyzed the centrosomal localization of CCP110 in WT and Luzp1[-/-] cells by immunofluorescence. Consistent with the higher ciliogenesis rate, CCP110 was present at two centrosomal spots at a lower proportion in Luzp1[-/-] cells (19%) compared to WT (84%; *Figure 6D and E*). This result suggests that the lack of LUZP1 might result in CCP110 reduction at the centrosome, leading to higher frequency of ciliogenesis in Luzp1[-/-] cells, and is reminiscent to the results obtained in TBS cells (*Bozal-Basterra et al., 2018*).

It is well-established that mammalian Shh signal transduction is dependent on functional primary cilia (*Huangfu et al., 2003*; *Yin et al., 2009*). Therefore, we examined whether Shh signaling is altered in Luzp1[-/-] cells. Cells were starved for 24 hr and incubated in the presence or absence of purmorphamine (a SMO agonist) for 24 hr to activate the Shh pathway. The mRNA expression of two Shh target genes (*Gli1* and *Ptch1*) was quantified by qRT-PCR (*Figure 7A,B*). We found that *Gli1* and *Ptch1* expression levels in non-treated Luzp1[-/-] cells were higher than in WT cells (*Gli1* 1.5 fold and *Ptch1* 2.3 fold increase in Luzp1[-/-] *vs* WT cells without purmorphamine) (*Figure 7A,B*). To further study the role of LUZP1 in Shh signaling, we analyzed GLI3 processing by western blot using total lysates extracted from WT *vs* Luzp1[-/-] cells. Without purmorphamine induction, we found a significantly higher ratio of GLI3 activating form *vs* GLI3 repressive form (GLI3-A:GLI3-R) in Luzp1[-/-] cells compared to WT (2.9 fold increase in Luzp1[-/-] cells *vs* WT) (*Figure 7C* and *Figure 7—figure*

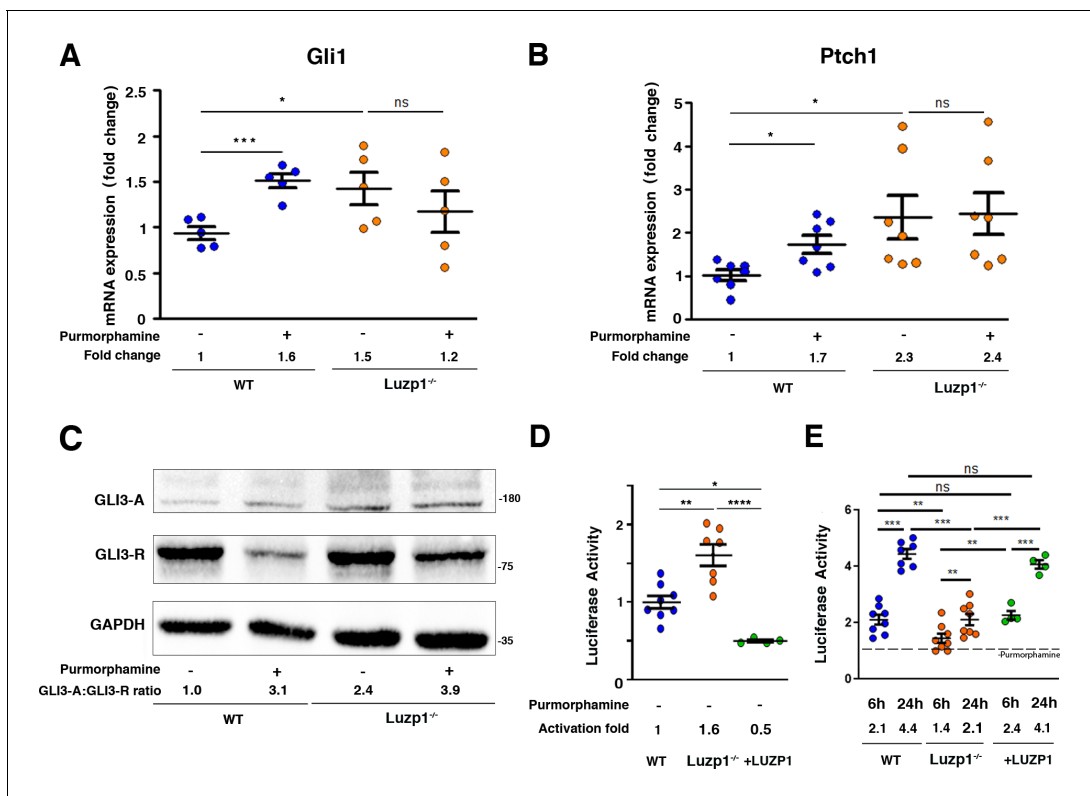

**Figure 7.** Luzp1[-/-] cells show aberrant Shh signaling. (**A, B**) Graphical representation of the fold-change in the expression of *Gli1* (n = 5) (**A**) and *Ptch1* (n = 7) (**B**) obtained by qRT-PCR from wild-type Shh-LIGHT2 cells (WT; blue dots) or Shh-LIGHT2 cells lacking *Luzp1* (Luzp1[-/-]; orange dots), treated (+) or not (-) with purmorphamine for 24 hr. (**C**) Western blot analysis of lysates from WT and Luzp1[-/-] cells. Samples were probed against GLI3 using an antibody that detects both GLI3-activator form (GLI3-A) and GLI3-repressor form (GLI3-R); GAPDH was used as loading control. Numbers under the lanes are the Mean of 3 independent experiments, resulting of dividing the activator by the repressor intensities, taking WT non-induced value as 1. Molecular weight markers are shown to the right. (**D, E**) Graphical representation of fold-change in luciferase activation when WT (n > 7; blue dots), Luzp1[-/-] (n > 7; orange dots) or +LUZP1 (n = 4; green dots) cells are treated for 6 and 24 hr or not (-) with purmorphamine. (**E**) Each treated condition was normalized against its respective non treated condition, taking the non-induced value as 1 (dashed line). All graphs represent the Mean and SEM. *P*-values were calculated using two-tailed unpaired Student´s t-test or One-way ANOVA and Bonferroni post-hoc test.

The online version of this article includes the following figure supplement(s) for figure 7:

**Figure supplement 1.** Full western blot images for *Figure 7*.

*supplement 1*). After induction, the values were similar for Luzp1⁻/⁻ and WT cells. We also examined the effects of lacking *Luzp1* on Shh signaling by measuring the activity of the Shh-responsive Firefly luciferase reporter in the presence or absence of purmorphamine for 24 hr to activate the Shh pathway (*Figure 7D,E*). Consistent with the *Gli1* and *Ptch1* qRT-PCR data, non-treated Luzp1⁻/⁻ cells showed higher Shh activity compared to control or +LUZP1 cells, as observed in TBS-derived cells (*Figure 7D*). However, the induction capacity of Luzp1⁻/⁻ cells upon purmorphamine treatment was reduced compared to WT or +LUZP1 cells (*Figure 7E*). Altogether, the observed defects in *Ptch1* and *Gli1* gene expression and Shh reporter misregulation point to a role for LUZP1 in Shh signaling.

## LUZP1 and F-actin levels are correlated

Based on the localization of LUZP1 to actin stress fibers and interaction with cytoskeletal proteins, we hypothesized that LUZP1 might also affect F-actin cytoskeleton. We observed a reduction in F-actin (labelled by phalloidin) in the Luzp1⁻/⁻ cells compared to WT, which was recovered in +LUZP1 cells (*Figure 8A*). We also note the correlation between LUZP1 levels and actin filaments in non-starved versus starved WT fibroblasts (*Figure 8B,C*). These results suggest that LUZP1 can stabilize actin stress fibers and that starvation triggers both LUZP1 and F-actin reduction.

## Truncated SALL1 promotes LUZP1 degradation via the ubiquitin proteasome system

In concordance with immunofluorescence results in *Figures 3* and *4*, we confirmed a reduction in total LUZP1 levels in TBS²⁷⁵ cells compared to controls by western blot (*Figure 9A,B* and *Figure 9—figure supplement 1*). No transcriptional changes in *LUZP1* expression were detected between

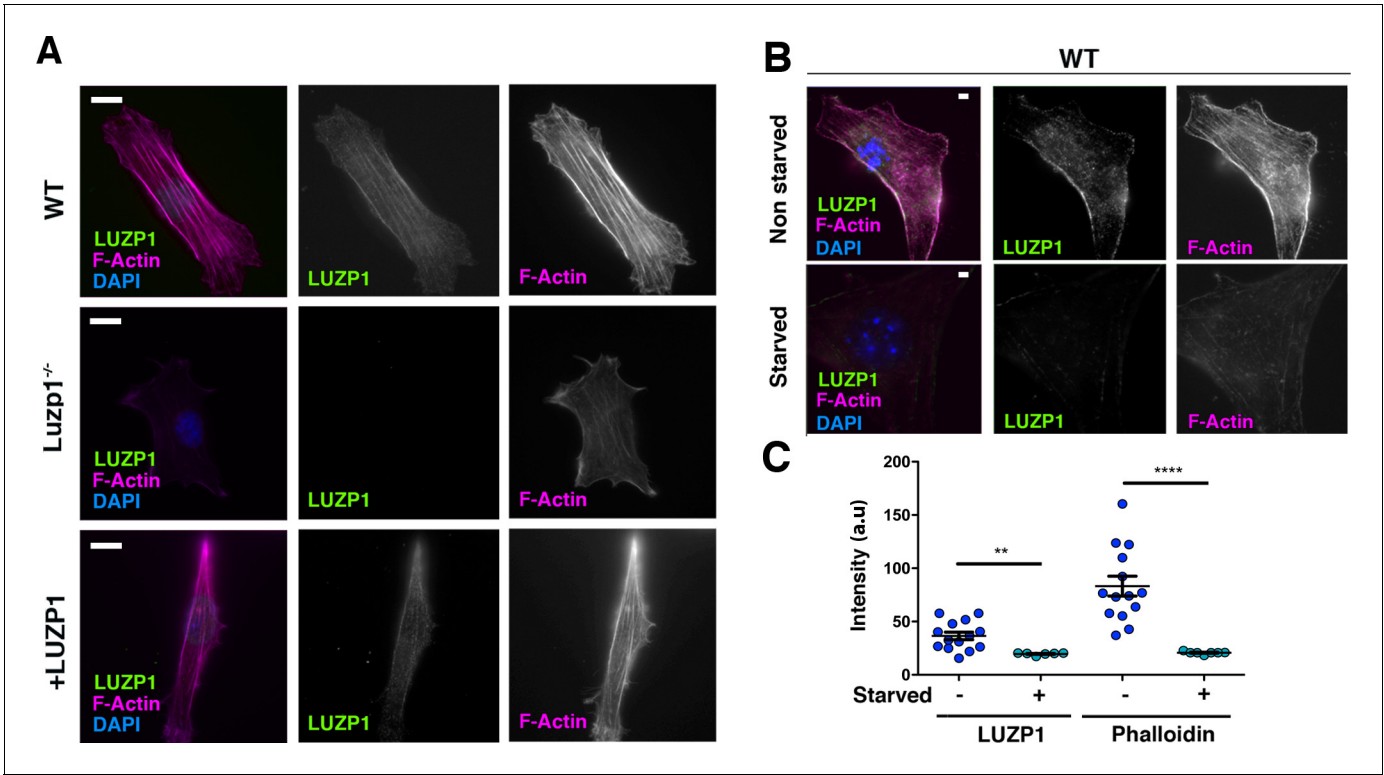

**Figure 8.** Cells missing Luzp1 show reduced F-actin levels. (**A**) Immunofluorescence micrographs of WT, Luzp1⁻/⁻ and +LUZP1 cells stained with an antibody against endogenous LUZP1 (green), phalloidin to detect F-actin (magenta), and counterstained with DAPI (blue). Single green and magenta channels are shown in black and white. Note the lack of LUZP1 signal in Luzp1⁻/⁻ cells. Scale bar, 10 µm. (**B**) Immunofluorescence micrographs of non-starved and starved WT cells stained with antibodies against endogenous LUZP1 (green), phalloidin (magenta) and DAPI (blue). Single green and magenta channels are shown in black and white. Scale bar, 5 µm. Imaging was performed using widefield fluorescence microscopy (Zeiss Axioimager D1, 63x objective). (**C**) Graphical representation of the LUZP1 or F-actin mean intensity (arbitrary units) as shown in (**B**). Graphs represent Mean and SEM of three independent experiments pooled together. *P*-values were calculated using One-way ANOVA and Bonferroni post-hoc test.

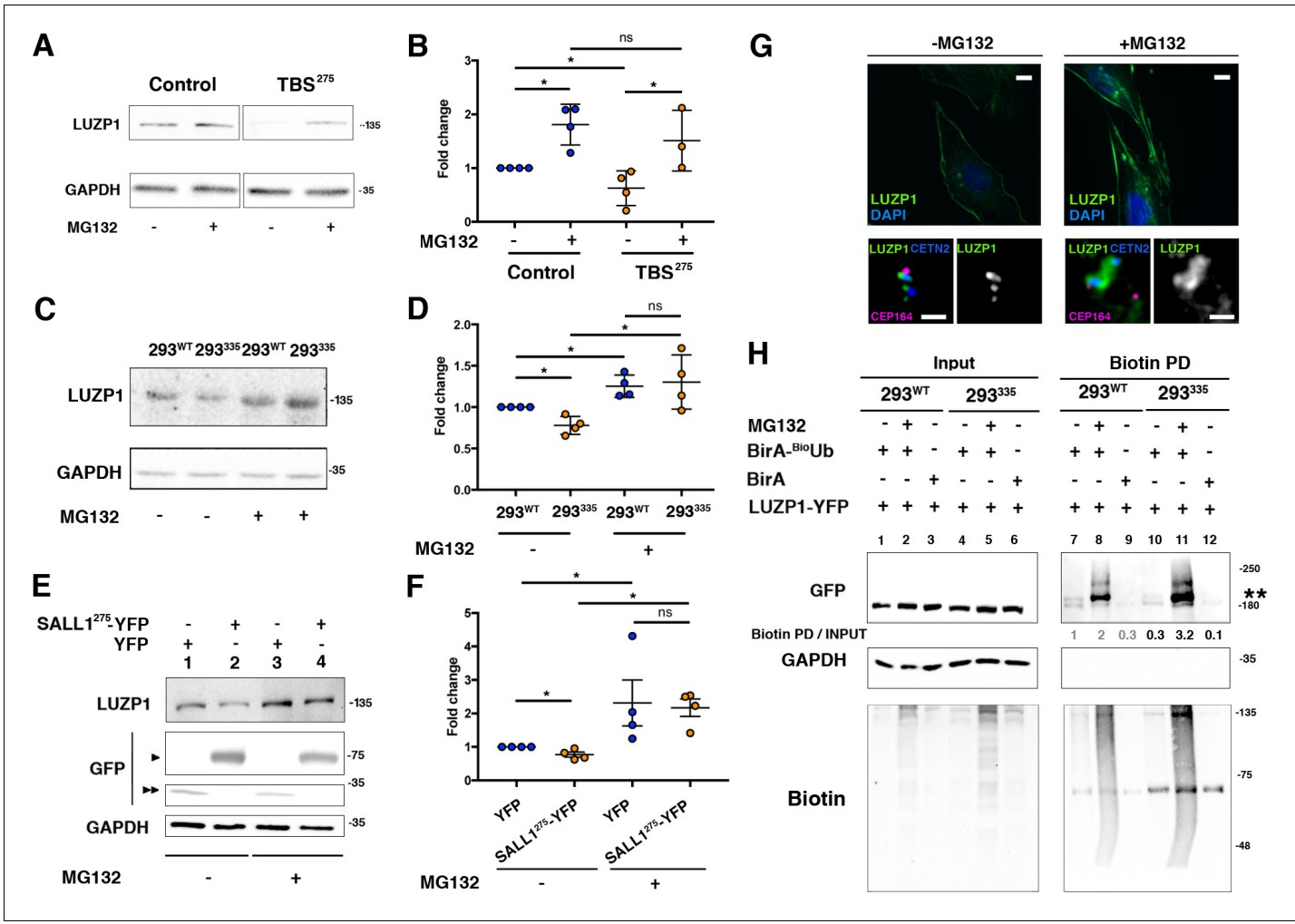

**Figure 9.** Truncated SALL1 leads to LUZP1 degradation through the UPS. (**A**) Representative western blot of Control and TBS[275] total cell lysates treated or not with MG132. A specific antibody detected endogenous LUZP1, and GAPDH was used as loading control. (**B**) Graphical representation of the fold changes of LUZP1/GAPDH ratios obtained in (**A**) for of Control (blue dots) and TBS[275] (orange dots) treated (+) or not (-) with the proteasome inhibitor MG132. Note the increase of LUZP1 until reaching control levels in TBS[275] cells upon MG132 treatment. (**C**) Representative western blot of 293[WT] and 293[335] total cell lysates treated or not with MG132. A specific antibody against LUZP1 detected endogenous LUZP1, and GAPDH was used as loading control. (**D**) Graphical representation of the fold changes of LUZP1/GAPDH ratios obtained in panel C for 293[WT] (blue dots) and 293[335] (orange dots) treated (+) or not (-) with MG132. Note that LUZP1 in 293[335] reaches control levels with MG132 treatment. (**E**) Representative western blot of total lysates of HEK 293FT cells transfected with *SALL1[275]-YFP* (lanes 1 and 3) or *YFP* alone (lanes 2 and 4) treated (+) or not (-) with MG132. Specific antibodies against LUZP1, GFP and GAPDH were used. One black arrowhead indicates SALL1[275]-YFP, two back arrowheads YFP alone. (**F**) Graphical representation of the fold changes of LUZP1/GAPDH ratios obtained in (**E**) for HEK 293FT cells transfected with *SALL1[275]-YFP* (orange dots) or *YFP* alone (blue dots) treated (+) or not (-) with MG132. Note that LUZP1 increases in the presence of MG132 when *SALL1[275]-YFP* was transfected. Data from at least three independent experiments pooled together are shown. *P*-values were calculated using two-tailed unpaired Student´s t-test. (**G**) Immunofluorescence micrographs of RPE1 cells treated (+MG132) or not (-MG132) with proteasome inhibitor showing LUZP1 associated with the cytoskeleton (upper panels) or in the centrosome (lower panels). Antibodies against endogenous LUZP1 (green), Centrin-2 (CETN2, blue) and CEP164 (magenta) were used. DAPI labelled the nuclei (blue). Single green channels are shown in black and white. Note the overall increase of LUZP1 upon MG132 treatment. Scale bar 10 µm (cytoskeleton panels) or 0.5 µm (centrosome panels). Imaging was performed using widefield fluorescence microscopy (Zeiss Axioimager D1, 63x objective). (**H**) Western blot analysis of input and biotin pulldown (PD) of HEK 293FT cells transfected with *CMV-LUZP1-YFP* and BioUb or BirA alone treated (+) or not (-) with MG132. Specific antibodies (GFP, GAPDH, Biotin) were used as indicated. Numbers under GFP panel are the result of dividing each biotin PD band intensity by the respective input band intensity and normalize them to lane 1. Molecular weight markers in kDa are shown to the right. Two asterisks indicate monoubiquitinated LUZP1.

The online version of this article includes the following figure supplement(s) for figure 9:

**Figure supplement 1.** Full western blot images for *Figure 9*.
**Figure supplement 2.** LUZP1 mRNA expression levels.

control and TBS[275] samples (*Figure 9—figure supplement 2*), so we hypothesized that truncated SALL1 might lead to UPS-mediated LUZP1 degradation. We analyzed LUZP1 levels after treatment with the proteasome inhibitor MG132, both in control and TBS[275] cells, and LUZP1 levels were increased to a higher extent in TBS[275] compared to control cells (1.8 fold increase in control *vs* 2.4 fold increase in TBS[275] cells) (*Figure 9A,B*). Moreover, we confirmed the reduction of LUZP1 levels in the CRISPR/Cas9 TBS model cell line (293[335]), compared to its parental cell line (293[WT]) (*Figure 9C,D* and *Figure 9—figure supplement 1*), and likewise in HEK 293FT cells stably overexpressing truncated SALL1 (*SALL1[275]-YFP*) compared to cells with *YFP* as control (*Figure 9E,F* and *Figure 9—figure supplement 1*). A more prominent increase in LUZP1 accumulation upon MG132 treatment was also observed in 293[335] and HEK 293FT cells overexpressing *SALL1[275]-YFP* compared to controls (*Figure 9C,D* and *Figure 9E,F*, respectively, and *Figure 9—figure supplement 1*). Additionally, we also observed LUZP1 accumulation upon MG132 treatment by immunofluorescence in RPE1 cells, both at the actin cytoskeleton (*Figure 9G*, upper panels) and at the centrosome (*Figure 9G*, lower panels). All together, these results show that LUZP1 levels are sensitive to degradation via the UPS pathway and suggest that truncated SALL1 may contribute to this process. Furthermore, we compared LUZP1 ubiquitination in 293[WT] *vs* 293[335] cells using the BioUb strategy (see Materials and methods) (*Pirone et al., 2017*). In the pulldowns, we could observe a prominent band in presence of BioUb, possibly corresponding to a monoubiquitinated form of LUZP1 (*Figure 9H* and *Figure 9—figure supplement 1*). This form was present in 293[WT] and 293[335] cells, and increased in the presence of MG132 in both cell lines. In addition, we observed a smear at higher molecular weight corresponding to polyubiquitinated forms of LUZP1 (*Figure 9H*, Biotin PD and *Figure 9—figure supplement 1*). Notably, the LUZP1 ubiquitinated pool relative to the input levels was higher in 293[335] compared to 293[WT] cells upon MG132 treatment (*Figure 9H*, Biotin PD, lane 8 *vs* lane 11 and *Figure 9—figure supplement 1*). These results suggest that truncated SALL1 promotes LUZP1 degradation through the UPS pathway.

## LUZP1 overexpression represses cilia formation and increases F-actin levels in TBS fibroblasts

Our results suggest that LUZP1 could be a mediator of TBS cilia phenotype and that this could be caused, at least in part, by the increased degradation of LUZP1 triggered by truncated SALL1. Therefore, increasing LUZP1 levels in TBS cells might affect the cilia and actin cytoskeleton phenotypes. To check whether an increase in LUZP1 levels is sufficient to repress ciliogenesis in primary human fibroblasts, Control and TBS[275] cells were transduced with *YFP* or *LUZP1-YFP* using lentivirus (*Figure 10A*). Whereas most non-transduced surrounding cells, as well as 100% of the TBS[275] cells expressing *YFP* were ciliated, only 40% of the Control and TBS[275] cells transduced with *LUZP1-YFP* displayed cilia (*Figure 10B*). Furthermore, we checked the actin cytoskeleton defects observed in TBS[275] cells to see the effect of overexpressing *LUZP1-YFP*. Immunostaining showed that *LUZP1-YFP* overexpression led to an increase in F-actin levels both in control and in TBS[275] cells compared to the surrounding non-transfected cells or TBS[275] cells overexpressing *YFP* (*Figure 10C*).

F-actin has a suppressive effect on ciliogenesis, and CytoD-mediated actin depolymerization has been shown to be permissive to cilia formation in cultured cells (*Kim et al., 2010*). To corroborate the relationship between LUZP1, actin and ciliogenesis, we performed CytoD treatment experiments. We observed that LUZP1-YFP overexpression can counteract the positive effects of CytoD on cilia formation (*Figure 10—figure supplement 1*). All together, these results support the notion that LUZP1 is a negative regulator of cilia formation and an F-actin stabilizing protein.

## Discussion

We conclude that LUZP1 might be a contributing factor to the TBS phenotype via its interaction with truncated SALL1 and its effect on ciliogenesis, based on several findings: i) LUZP1 levels are reduced in TBS-derived cells likely due to truncated SALL1-mediated degradation through the UPS; ii) LUZP1 interacts with proteins of the centrosome and of the actin cytoskeleton; iii) In the absence of LUZP1, the assembly and growth of primary cilia is enhanced in cycling cells, accompanied by an increase in basal Shh signaling, and the actin cytoskeleton is reduced; iv) Increase of LUZP1 reduces the levels of ciliogenesis in TBS-individuals derived cells. All these results do not rule out the possibility that, in addition to LUZP1, other factors might be contributing to TBS etiology.

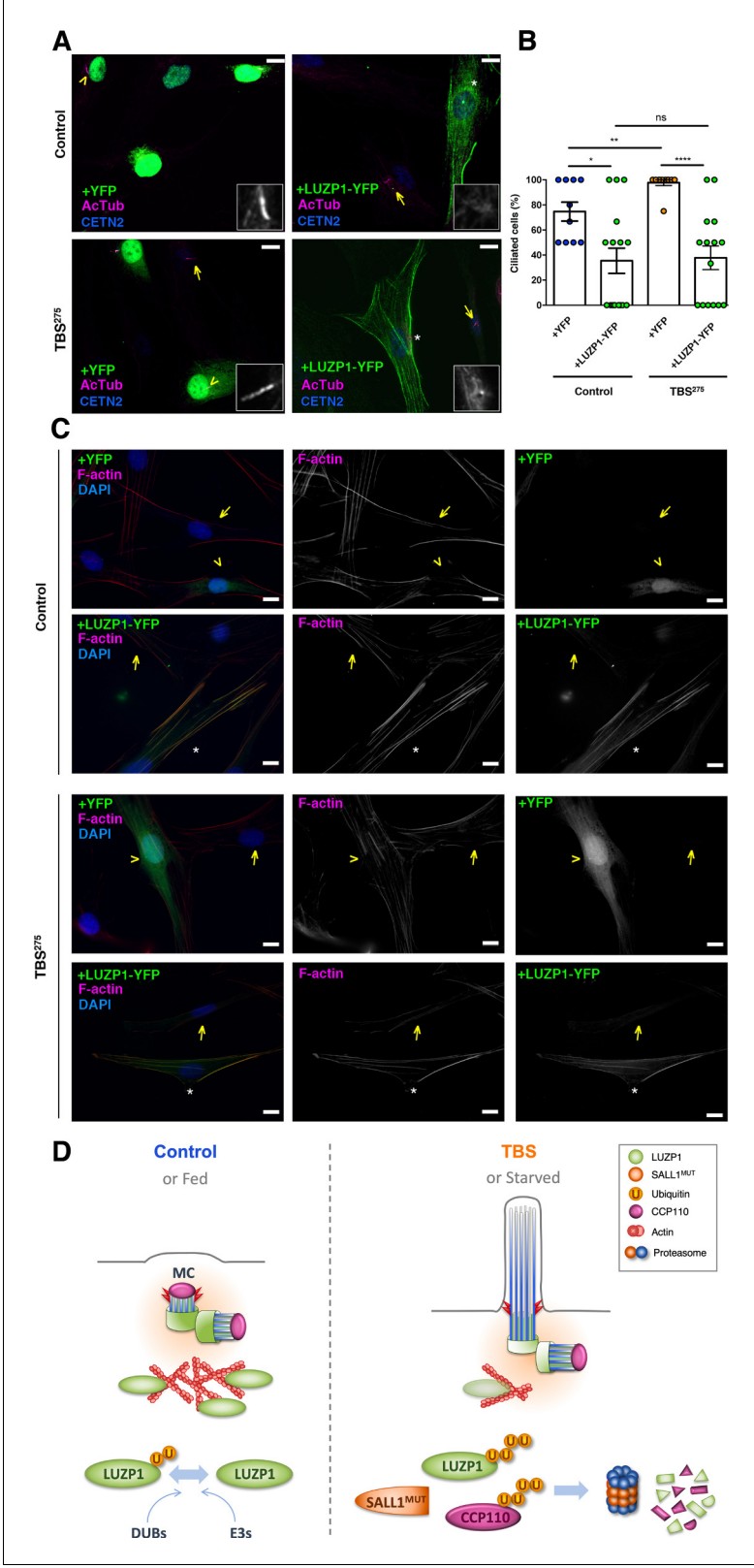

**Figure 10.** LUZP1 overexpression suppresses ciliogenesis and increases F-actin levels. (**A**) Representative micrographs of ciliated Control and TBS[275] cells transduced with *YFP* or *LUZP1-YFP* expressing lentivirus. Yellow arrowhead and white asterisk point at a magnified region shown in the lower right panel in black and white. Note the lack of cilia in cells transduced with LUZP1-YFP (white asterisk) compared to non-transduced cells (yellow

*Figure 10 continued on next page*

Figure 10 continued

arrow). AcTub: acetylated alpha-tubulin (magenta); CETN2: Centrin-2 (blue). (B) Graphical representation of the number of ciliated cells per micrograph in Control and TBS[275] cells overexpressing *YFP* or *LUZP1-YFP* (n > 10 micrographs). Graphs represent Mean and SEM of ciliation frequencies per micrograph in three independent experiments pulled together. *P*-values were calculated using One-way ANOVA and Bonferroni post-hoc test or two-tailed unpaired Student´s t-test. (C) Representative micrographs of Control and TBS[275] cells transduced with *YFP* (yellow arrowhead) or *LUZP1-YFP* (white asterisk) co-stained with phalloidin to label F-actin (magenta) and DAPI (blue). Note the increase in F-actin levels in cells transduced with LUZP1-YFP (white asterisk) compared to non-transfected cells (yellow arrow). Scale bar, 10 µm. Imaging was performed using widefield fluorescence microscopy (Zeiss Axioimager D1, 63x objective). (D) Schematic model representing how the presence of truncated SALL1 might cause cilia and actin malformations in TBS through LUZP1 interaction and UPS-mediated degradation. In control (or fed) cells (left), LUZP1 (in green) localizes to F-actin and to the proximal ends of the two centrioles, inhibiting cilia formation. LUZP1 proteostasis is maintained by the Ubiquitin Proteasome System (UPS). By contrast, in TBS (or starved) cells (right) the truncated form of SALL1 interacts with LUZP1 and promotes its UPS-mediated degradation. Likewise, others have shown ubiquitination and proteasome-mediated degradation of CCP110 at the mother centriole (MC) is permissive for ciliogenesis (*Li et al., 2013*). LUZP1 reduction at the centrosome and at the cytoskeleton favors the formation of the primary cilia.

The online version of this article includes the following figure supplement(s) for figure 10:

**Figure supplement 1.** Exogenous LUZP1 expression suppresses the positive effects of CytoD on ciliogenesis.

## LUZP1 localizes to the centrosome and actin cytoskeleton

LUZP1 was previously described as a nuclear protein, with expression limited to the mouse brain (*Lee et al., 2001*; *Sun et al., 1996*). We tested two different commercial antibodies against LUZP1 by immunofluorescence and, while nuclear localization was sporadic and weak, the most prominent localization of LUZP1 was observed in the actin cytoskeleton and centrosome, both in human and mouse cells. This localization is consistent with our TurboID analysis that showed an enrichment of factors associated with the actin cytoskeleton and/or centrosomes among the potential interactors of LUZP1. The localization of LUZP1 to the actin cytoskeleton, as well as its expression in tissues beyond the brain, is consistent with independent validation in cell lines by the Human Protein Atlas (HPA; proteinatlas.org) and other expression databases (e.g. EMBL EBI Expression Atlas ebi.ac.uk/gxa). Two independent proximity labeling studies identified LUZP1 as a proximal interactor of centriole (*Gupta et al., 2015*) and centriolar satellite-related proteins (*Gheiratmand et al., 2019*). Moreover, LUZP1 has been recently reported as a centrosomal protein involved in cilia regulation (*Gonçalves et al., 2019*). These localizations are also consistent with fluorescent LUZP1 fusion proteins (this work; [*Gupta et al., 2015*; *Gonçalves et al., 2019*] #117). Discrepancies with the previously reported LUZP1 localization and distribution might be due to technical differences, such as epitope specificity for the antisera used in the immunohistochemistry.

Here, we report that LUZP1 surrounds the proximal end of both centrioles. Like LUZP1, a large number of centrosomal scaffold proteins contain coiled-coil regions, and the proteins are concentrically localized around a centriole in a highly organized fashion (e.g. Cep120, Cep57, Cep63, Cep152, CPAP, Cdk5Rap2, PCNT) (*Fu and Glover, 2012*; *Lawo et al., 2012*; *Mennella et al., 2012*). Furthermore, we show that LUZP1 interacts with centrosome and actin-related proteins (*Figure 3* and *Figure 4*). LUZP1 is associated with CCP110 and CEP97, using pull down, immunoprecipitation and proximity proteomics approaches. This association is likely to be dynamic and potentially indirect, via other bridging factors, since our immunostainings show that LUZP1 and CCP110 are located in different areas of the centrioles. LUZP1 has also been identified as an interactor of ACTR2 (ARP2 actin related protein two homologue) and FLNA (*Hein et al., 2015*; *Wang and Nakamura, 2019*), and it has been recently described as an actin cross-linking protein (*Wang and Nakamura, 2019*). We found that LUZP1 localizes not only to centrioles and actin cytoskeleton, but also to the midbody in dividing cells, which was recently reported to influence ciliogenesis in polarized epithelial cells (*Bernabé-Rubio et al., 2016*). Our data suggest that the association of LUZP1 to centrosomes and actin filaments may contribute to its overall roles.

## LUZP1 as an integrator of actin and primary-cilium dynamics

Actin dynamics coordinate several processes that are crucial for ciliogenesis. For example, positioning the MC to the appropriate area at the cell cortex is an actin-dependent process (*Boisvieux-Ulrich et al., 1990*; *Euteneuer and Schliwa, 1985*). A reduction in cortical actin should promote ciliogenesis, since there is less physical restriction for docking of the MC. Supporting this hypothesis, several studies have found that disruptions in the actin cytoskeleton, induced either chemically or genetically, promote ciliogenesis or affect cilia length (*Cao et al., 2012*; *Drummond et al., 2018*; *Hernandez-Hernandez et al., 2013*; *Kang et al., 2015*; *Kim et al., 2015*; *Kim et al., 2010*). How actin regulates cilium length is not clear. Recently, a role for actin has been implicated in ectocytosis and cilium tip scission, preventing the axoneme from growing too long (*Nager et al., 2017*; *Phua et al., 2017*). Whether LUZP1 at the centrosome is complexed with filamin and actin is unknown, but if so, they could together serve to stabilize the basal body as the axoneme extends or is subjected to mechanical stress.

## LUZP1 is altered in TBS-derived cells

TBS is caused by mutations in *SALL1* gene, which give rise to truncated proteins that interfere with the normal function of the cell. We show that LUZP1interacts with truncated SALL1 and with SALL1$^{FL}$, suggesting that interaction occurs through an N-terminal domain shared by both. In control cells, LUZP1 and SALL1$^{FL}$ likely have minimal or no interaction due to their respective localizations to the cytoplasm and nucleus. However, truncated SALL1, alone or together with SALL1$^{FL}$ that is retained in the cytoplasm, can interact with cytoplasmic LUZP1, promoting its degradation and functional inhibition. Importantly, we detected an increase in LUZP1 levels upon treatment with the proteasome inhibitor MG132 (*Figure 9*), suggesting that LUZP1 degradation is proteasome-dependent. Next, we demonstrated that LUZP1 is ubiquitinated, and that truncated SALL1 both increases LUZP1 ubiquitination and decreases its stability. In agreement with our findings, LUZP1 ubiquitination has been detected in several proteomic screens for ubiquitinated proteins (*Akimov et al., 2018*; *Mertins et al., 2013*; *Povlsen et al., 2012*; *Udeshi et al., 2013*; *Wagner et al., 2012*).

Further experiments would be required to understand the precise mechanism by which truncated SALL1 can influence LUZP1 ubiquitination, but one possibility could be de novo complexes involving specific Ub E3 ligases or de-ubiquitinases which could influence LUZP1 stability. In fact, various E3s/de-ubiquitinases, as well as other components of the UPS, were found as proximal interactors of truncated SALL1 and LUZP1. Furthermore, regulation by the UPS system has been reported for centrosomal factors, including the cilia regulator CCP110 (*D'Angiolella et al., 2010*; *Hossain et al., 2017*; *Li et al., 2013*).

Both Luzp1$^{-/-}$ and TBS cells showed a reduction in F-actin accompanied by an increase in ciliation. We suggest that the reduction in F-actin in TBS cells might contribute to their higher cilia abundance, longer cilia and increased Shh signaling. An increase in LUZP1 in control and TBS$^{275}$ cells is sufficient to increase F-actin levels and reduce cilia frequency, supporting that LUZP1 may have a contributing role in the TBS phenotype.

## The role of LUZP1 in TBS phenotype

Cilia formation, Shh signaling, and the actin cytoskeleton is aberrant in TBS patient-derived fibroblasts (this work; [*Bozal-Basterra et al., 2018*]). Changes in Shh signaling have not been reported in mouse models for TBS, nor in any other cell types or tissues derived from TBS patients. Nevertheless, the phenotypes observed in TBS individuals fall within the spectrum of those observed in ciliopathies, characterized by malformations in digits, ears, heart, brain, kidneys and urogenital anomalies, phenotypes that are consistent with misregulated Shh signaling. Preaxial polydactyly has been associated with ectopic Shh expression in limbs (*Dunn et al., 2011*; *Johnson et al., 2014*; *Lettice et al., 2003*; *Lettice et al., 2008*; *Liem et al., 2009*; *Zhulyn and Hui, 2015*); Anal stenosis or imperforate anus have been related to misregulation of Shh pathway (*Kang et al., 1997*; *Mo et al., 2001*; *Roberts et al., 1995*), as well as deafness and dysplastic ears (*Driver et al., 2008*).

We observed primary cilia, Shh signaling and cytoskeletal defects in Luzp1$^{-/-}$ cells. Several studies have implicated defective primary cilia and Shh signaling in the etiology of neural tube closure defects, as well as crucial roles for the actin cytoskeleton (*Wallingford, 2005*). There are certain parallels between phenotypes observed in animal models of Luzp1 and Sall1. Exencephaly and neural

tube defects were detected in mice and *Xenopus Sall1* mutants (*Böhm et al., 2008*; *Exner et al., 2017*; *Kiefer et al., 2003*). Luzp1 KO mice exhibit ectopic Shh expression in the hindbrain neuroepithelium and display NTDs, however the expression of Shh-responsive genes (such as Gli1 or Ptch1) was not reported (*Hsu et al., 2008*). Perhaps the role of LUZP1 in Shh signaling, in spatial control of the signal or the response (or both), contributes to the NTD phenotype. Exencephaly may also be caused by failure in bending at the dorsolateral hinge point of the neural folds, where cells undergo changes in apical actin architecture (*Sadler et al., 1982*). *Luzp1 KO* embryos exhibited dorsolateral neural folds that were convex instead of the concave morphology observed in WT embryos (*Hsu et al., 2008*), suggested that defective actin dynamics may contribute to the NTD phenotype. Taken together, defective actin dynamics, aberrant primary cilia and changes in Shh signaling might lead to NTDs observed in the LUZP1 mouse model, as well as other animal models of TBS and loss of Sall-related proteins.

In addition to NTDs, cardiac malformations are another feature found in human ciliopathies (*Klena et al., 2017*). Cardiac defects are observed in *Luzp1* knockout mice (*Hsu et al., 2008*), as well as TBS patients (*Kohlhase, 1993*). TBS cardiac defects include atrial or ventricular septal defect, the latter of which is seen in *Luzp1* knockout mice. Moreover, compound *Sall1/Sall4* KO mutant mice exhibit both NTDs and cardiac problems (*Böhm et al., 2008*). While *Luzp1* and *Sall1* may both contribute to neural tube and heart development, a novel crosstalk may arise in TBS due to dominantly-acting truncated SALL1 that could derail these processes and cause deformities.

In conclusion, our data indicate that LUZP1 functions as a cilia suppressor (*Figure 10D*). It localizes to actin stress fibers and to the centrosome. In starved cells, likewise in TBS cells, overall LUZP1 levels are diminished in both structures, which facilitates the formation of the primary cilia. In TBS cells, the truncated form of SALL1 localizes to the cytoplasm, interacting with LUZP1 and other factors, leading to the degradation of LUZP1, simulating what happens when control cells undergo starvation. As a result, the frequency of cilia formation increases, and cilia are longer than in control cells.

Ciliogenesis requires communication between the actin cytoskeleton and the centrosome. Here, we propose that LUZP1 might contribute as a nexus in this complex intracellular network that is disrupted by truncated SALL1. Our findings point to the intriguing possibility that LUZP1 might be a key relay switch in this network that, together with other factors, might contribute to the phenotypes observed in TBS.

## Materials and methods

### Cell culture

TBS-derived primary fibroblasts, U2OS (ATCC HTB-96), HEK 293FT (Invitrogen R70007), and mouse Shh-LIGHT2 cells (*Taipale et al., 2000*) were cultured at 37°C and 5% $CO_2$ in Dulbecco's modified Eagle medium (DMEM) supplemented with 10% fetal bovine serum (FBS, Gibco) and 1% penicillin/streptomycin (Gibco). Human telomerase reverse transcriptase immortalized retinal pigment epithelial cells (hTERT-RPE1, ATCC CRL-4000; designated here as RPE1) were cultured in DMEM:F12 (Gibco) supplemented with 10% FBS and 1% penicillin and streptomycin. Dermal fibroblasts carrying the *SALL1* pathogenic variant c.826C > T (*SALL1[c.826C>T]*), that produce a truncated protein p. Leu275* (SALL1[275]), were derived from a male TBS individual UKTBS#3 (called here TBS[275]) (*Bozal-Basterra et al., 2018*). Adult female dermal fibroblasts (ESCTRL#2) from healthy donors were used as controls. Cell lines were authenticated by commercial providers (Invitrogen, ATCC). Additional validation was done by TBS allele genotyping (primary fibroblasts) and reporter selection (zeoR, neoR; Shh-LIGHT2). Cultured cells were maintained between 10 and 20 passages, tested for senescence by γ-H2AX staining and mycoplasma contamination and grown until confluence (6-well plates for RNA extraction and western blot assays; 10 cm dishes for pulldowns). The use of human samples in this study was approved by the institutional review board (Ethics Committee at CIC bioGUNE, protocol P-CBG-CBBA-2111) and appropriate informed consent was obtained from human subjects or their parents.

## Cell synchronization and drug treatment

RPE1 cells were arrested in G1 phase by treatment with mimosine (Sigma, 400 µM) for 24 hr. For S phase arrest, cells were subjected to thymidine treatment (Sigma, 2.5 mM) for 16 hr, followed by release for 8 hr, and subsequently blocked again for 16 hr. For G2/M phase arrest, cells were treated with RO-3306 (Sigma, 10 µM) for 20 hr. For G0 phase synchronization and inducing primary cilia formation, cells were starved for 48 hr (DMEM, 0% FBS, 1% penicillin and streptomycin). Treatments with the proteasome inhibitor MG132 (Calbiochem, 5 µM) were for 15 hr and with CytoD (Sigma, 50 nM) for 16 hr to stimulate actin depolymerization. HEK 293FT cells were transfected using calcium phosphate method and U2OS cells using Effectene Transfection Reagent (Qiagen).

## CRISPR-Cas9 genome editing

CRISPR-Cas9 targeting of *SALL1* locus was performed to generate a HEK 293FT cell line carrying a TBS-like allele (*Bozal-Basterra et al., 2018*). The mouse *Luzp1* locus was targeted in NIH3T3-based Shh-LIGHT2 fibroblasts (*Taipale et al., 2000*) (kind gift of A. McGee, Imperial College). These are NIH3T3 mouse fibroblasts that carry an incorporated Shh reporter (firefly luciferase under control of *Gli3*-responsive promoter). Cas9 was introduced into Shh-LIGHT2 cells by lentiviral transduction (Lenti-Cas9-blast; Addgene #52962; kind gift of F. Zhang, MIT) and selection with blasticidin (5 µg/ml). Two high-scoring sgRNAs were selected (http://crispr.mit.edu/) to target near the initiation codon (sg2: 5′-CTTAAATCGCAGGTGGCGGT_TGG-3′; sg3: 5′-CTTCAATCTTCAGTACCCGC_TGG-3′). These sequences were cloned into px459 2.0 (Addgene #62988; kind gift of F. Zhang, MIT), for expressing both sgRNAs and additional Cas9 with puromycin selection. Transfections were performed in Shh-LIGHT2/Cas9 cells with Lipofectamine 3000 (Thermo). 24 hr after transfection, transient puromycin selection (0.5 µg/ml) was applied for 48 hr to enrich for transfected cells. Cells were plated at clonal density, and well-isolated clones were picked and propagated individually. Western blotting was used to identify clones lacking *Luzp1* expression. Further propagation of a selected clone (#6) was carried out with G418 (0.4 mg/ml) and zeocin (0.15 mg/ml) selection to maintain expression of luciferase reporters. Genotyping was performed using genomic PCR (*MmLuzp1_geno_for*: 5′-GTTGCCAAAGAAGGTTGTGGATGCC-3′; *MmLuzp1_geno_rev*: 5′-CGTAAGGTTTTCTTCCTCTTCAAGTTTCTC-3′). We found that Luzp1[-/-] cells presented a homozygous deletion of the sequence: 5′-ccacctgcgatttaagttacagagcctgagccgccgcctcgatgagttagaggaagctacaaaaaacctccagagagcagaggatgagctcctggacctccaggacaaggtgatccaggcagagggcagcgactccagcacgctggctgagatcgaggtgctgcgccagcgg-3′. This generated a truncation and a stop codon early in the N-terminal part of the protein. The resulting peptide was: MAELTNYKDAASNRY*. A rescue cell line was generated by transducing Shh-LIGHT2 *Luzp1* KO clone #6 with a lentiviral expression vector carrying EFS-LUZP1-YFP-P2A-blast[R], with a positive population selected by fluorescence-activated cell sorting.

## Plasmid construction

*SALL1* truncated (*SALL1[275]-YFP or Myc-BirA*-SALL1[275]*) and FL versions (*SALL1[FL]-YFP*, *SALL1[FL]-2xHA or Myc-BirA*-SALL1[FL]*) were previously described (*Bozal-Basterra et al., 2018*). Human *LUZP1* ORF was amplified by high-fidelity PCR (Platinum SuperFi; Thermo) from RPE1 cell cDNA and cloned to generate *CB6-GFP-LUZP1*. This was used as a source clone to generate additional variants, including *CMV-LUZP1-YFP*. The LUZP1-YFP and TbID-LUZP1 lentiviral expression vectors were generated by replacing Cas9 in Lenti-Cas9-blast (Addgene #52962). All constructs were verified by Sanger sequencing. Plasmids *CAG-BioUBC(x4)_BirA_V5_puro* (called here BioUb) and *CAG-BirA-puro* (called here BirA) were reported previously (*Pirone et al., 2017*).

## Biotin pulldowns

Using the BioID and the TurboID methods (*Branon et al., 2018*; *Roux et al., 2012*), proteins in close proximity to SALL1 and LUZP1, respectively, were biotinylated and isolated by streptavidin-bead pulldowns. For transient transfections, *Myc-BirA*-SALL1[c.826C>T]* or *Myc-BirA*-SALL1FL* were used in HEK 293FT cells (10 cm dishes). For TurboID experiments, *TbID-LUZP1-P2A-blast* or *TbID-P2A-blast* alone were transduced in RPE1 cells and a stable population was selected. For the isolation of BioUb-conjugates 10 cm dishes were transfected with BioUb or BirA as control (*Pirone et al., 2017*). Briefly, 24 hr after transfection, medium was supplemented with biotin at 50 µM. Cells were collected 48 hr after transfection, washed 3 times on ice with cold phosphate buffered saline (PBS) and

scraped in lysis buffer [8 M urea, 1% SDS, 1x protease inhibitor cocktail (Roche), 60 μM NEM in 1x PBS; 1 ml per 10 cm dish]. At room temperature, samples were sonicated and cleared by centrifugation. Cell lysates were incubated overnight with 40 μl of equilibrated NeutrAvidin-agarose beads (Thermo Scientific). Beads were subjected to stringent washes using the following washing buffers (WB), all prepared in PBS: WB1 (8 M urea, 0.25% SDS); WB2 (6 M Guanidine-HCl); WB3 (6.4 M urea, 1 M NaCl, 0.2% SDS), WB4 (4 M urea, 1 M NaCl, 10% isopropanol, 10% ethanol and 0.2% SDS); WB5 (8 M urea, 1% SDS); and WB6 (2% SDS). For elution of biotinylated proteins, beads were heated at 99°C in 50 μl of Elution Buffer (4x Laemmli buffer, 100 mM DTT). Beads were separated by centrifugation (18000 x g, 5 min).

## Lentiviral transduction

Lentiviral expression constructs were packaged using psPAX2 and pVSV-G (Addgene) in HEK 293FT cells, and lentiviral supernatants were used to transduce Shh-LIGHT2 cells, RPE1 cells, or TBS[275] and control human fibroblasts. Stable-expressing populations were selected using puromycin (1 μg/ml) or blasticidin (5 μg/ml). The vectors EFS-LUZP1-YFP-P2A-blast[R], EFS-YFP-P2A-blast[R], LL-GFS-SALL1[c.826C>T]-IRES-puro[R], LL-GFS-stop-IRES-puro[R], EFS-TbID-LUZP1-P2A-blast[R] and EFS-TbID-P2A-blast[R] were used. Lentiviral supernatants were concentrated 100-fold before use (Lenti-X concentrator, Clontech). Concentrated virus was used for transducing primary fibroblasts and RPE1 cells.

## Mass spectrometry

Analysis was done in RPE1 cells stably expressing TbID or TbID-LUZP1 at sub-endogenous levels. Three independent pulldown experiments ($1.5 \times 10^8$ cells per replicate) were analyzed by MS. Samples eluted from the NeutrAvidin beads were separated in SDS-PAGE (50% loaded) and stained with Sypro-Ruby (Biorad) according to manufacturer's instructions. Entire gel lanes were excised, divided into pieces and in-gel digested with trypsin. Recovered peptides were desalted using stage-tip C18 microcolumns (Zip-tip, Millipore) and resuspended in 0.1% FA prior to MS analysis. Samples were analyzed in a novel hybrid trapped ion mobility spectrometry – quadrupole time of flight mass spectrometer (timsTOF Pro with PASEF, Bruker Daltonics) coupled online to a nanoElute liquid chromatograph (Bruker). This mass spectrometer takes advantage of a novel scan mode termed parallel accumulation – serial fragmentation (PASEF), which multiplies the sequencing speed without any loss in sensitivity (*Meier et al., 2015*), providing outstanding analytical speed and sensibility for proteomics analyses (*Meier et al., 2018*). Sample (200 ng) was directly loaded in a 15 cm Bruker nanoelute FIFTEEN C18 analytical column (Bruker) and resolved at 400 nl/min with a 100 min gradient. Column was heated to 50°C using an oven. Protein identification and quantification was carried out using PEAKS software (Bioinformatics solutions). Searches were carried out against a database consisting of human entries (Uniprot/Swissprot), with precursor and fragment tolerances of 20 ppm and 0.05 Da. Only proteins identified with at least two peptides at FDR < 5% were considered for further analysis. Data were loaded onto Perseus platform (*Tyanova et al., 2016*) and further processed ($Log_2$ transformation, selection of proteins with at least two valid values in at least one condition, imputation). A t-test was applied in order to determine the statistical significance of the differences detected, and heatmaps were generated using this tool. Protein IDs were ranked according to the number of peptides found and their corresponding intensities. Gene ontology (GO) term enrichment was analyzed using g:GOSt Profiler, a tool integrated in the g:Profiler web server (*Reimand et al., 2016*). GO enrichment was obtained by calculating $-Log_{10}$ of the P-value.

Network analysis of LUZP1 interactors was performed using the String app version 1.4.2 in Cytoscape version 3.7.2, with a high confidence interaction score (0.7). Transparency and width of the edges were continuously mapped to the String score (textmining, databases, coexpression, experiments, fusion, neighborhood and cooccurrence). Color, transparency and size of the nodes were discretely mapped to the Log2 enrichment value as described in *Figure 1*. The Molecular COmplex DEtection (MCODE) plug-in version 1.5.1 was used to identify highly connected subclusters of proteins (degree cutoff of 2; Cluster finding: Haircut; Node score cutoff of 0.5; K-Core of 2; Max. Depth of 100).

## GFP-trap pulldowns

All steps were performed at 4°C. HEK 293FT transfected cells were collected after 48 hr, washed 3 times with 1x PBS and lysed in 1 ml of lysis buffer [25 mM Tris-HCl pH 7.5, 150 mM NaCl, 1 mM EDTA, 1% NP-40, 0.5% Triton X-100, 5% glycerol, protease inhibitors (Roche)]. Lysates were kept on ice for 30 min vortexing every 5 min and spun down at 25,000 x g for 20 min. After saving 40 µl of supernatant (input), the rest was incubated overnight with 30 µl of pre-washed GFP-Trap resin (Chromotek) in a rotating wheel. Beads were washed 5 times for 5 min each with WB (25 mM Tris-HCl pH 7.5, 300 mM NaCl, 1 mM EDTA, 1% NP-40, 0.5% TX-100, 5% glycerol). Beads were centrifuged at 2000 x g for 2 min after each wash. For elution, samples were boiled for 5 min at 95°C in 2x Laemmli buffer.

## Immunoprecipitation

All steps were performed at 4°C. Cells were collected and lysates were processed as described for GFP-trap pulldowns. After saving 40 µl of supernatant (input), the rest was incubated overnight with 1 µg of anti-CEP97 antibody (Proteintech), or anti-LUZP1 antibody (Sigma HPA028506) and for additional 4 hr with 40 µl of pre-washed Protein G Sepharose 4 Fast Flow beads (GE Healthcare) on a rotating wheel. Beads were washed 5 times for 5 min each with WB (10 mM Tris-HCl pH 7.5, 137 mM NaCl, 1 mM EDTA, 1% Triton X-100). Beads were centrifuged at 2000 x g for 2 min after each wash. For elution, samples were boiled for 5 min at 95°C in 2x Laemmli buffer.

## Western blot analysis

Cells were lysed in cold RIPA buffer (Cell Signaling Technology), WB5 (8 M urea, 1% SDS) or weak buffer (10 mM PIPES pH 6.8, 100 mM NaCl, 1 mM EGTA, 3 mM MgCl2, 300 mM sucrose, 0.5 mM DTT, 1% Triton X-100) supplemented with 1x protease/phosphatase inhibitor cocktail (Roche). Lysates were kept on ice for 30 min vortexing every 5 min and then cleared by centrifugation (25,000 x g, 20 min, 4°C). Supernatants were collected and protein content was quantified by BCA protein quantification assay (Pierce). After SDS-PAGE and transfer to nitrocellulose membranes, blocking was performed in 5% milk, or in 5% BSA (Bovine Serum Albumin, Fraction V, Sigma) in PBT (1x PBS, 0.1% Tween-20). In general, primary antibodies were incubated overnight at 4°C and secondary antibodies for 1 hr at room temperature (RT). Antibodies used: rabbit anti-LUZP1 (Proteintech, 1:1,000) for *Figure 1* and rabbit anti-LUZP1 (Sigma HPA028506, 1:1,000) for the rest of the experiments, rabbit anti-CCP110 (Proteintech, 1:1,000), rabbit anti-CEP97 (Proteintech, 1:1,000), mouse anti-GFP (Roche, 1:1,000), mouse anti-GAPDH (Proteintech, 1:1,000), mouse anti-FLNA (Merck, 1:1,000), rabbit anti-BirA (Sino Biological, 1:1000), HRP-conjugated anti-biotin (Cell Signaling Technology, 1:2,000), rabbit anti Myc (Cell Signaling Technology, 1:2,000), mouse anti-actin (Sigma, 1:1,000), goat anti-GLI3 (R and D, 1:1,000) and mouse anti-SALL1 (R and D, 1:1,000). Secondary antibodies were anti-mouse or anti-rabbit HRP-conjugates (Jackson Immunoresearch). Proteins were detected using Clarity ECL (BioRad) or Super Signal West Femto (Pierce). Quantification of bands was performed using ImageJ software and normalized against GAPDH or actin levels. At least three independent blots were quantified per experiment.

## Immunostaining

Shh-LIGHT2 cells, RPE1, U2OS cells and primary fibroblasts from control and TBS individuals were seeded on 11 mm coverslips (15,000–25,000 cells per well; 24well plate). After washing 3 times with cold 1xPBS, cells were fixed with 100% methanol for 10 min at −20°C or with 4% PFA supplemented with 0.1% Triton X-100 in PBS for 15 min at RT. Then, coverslips were washed 3 times with 1x PBS. Blocking was performed for 1 hr at 37°C in blocking buffer (BB: 2% fetal calf serum, 1% BSA in 1x PBS). Primary antibodies were incubated overnight at 4°C and cells were washed with 1x PBS 3 times. To label the ciliary axoneme and the basal body/pericentriolar region, we used mouse antibodies against acetylated alpha-tubulin (Santa Cruz Biotechnologies, 1:160) and gamma-tubulin (Proteintech, 1:160). Other antibodies include: rat anti-Centrin-2 (CETN2, Biolegend, 1:160), rabbit anti-LUZP1 (Sigma HPA028506, 1:100), rabbit anti-CCP110 (Proteintech, 1:200), rabbit anti-PCM1 (Cell Signaling Technology, 1:100), rabbit anti ODF2 (Atlas, 1:100), mouse anti-CEP164 (Genetex, 1:100), mouse anti beta-tubulin (DSHB, 1:100) and pericentrin (Abcam, 1:100).

Donkey anti-rat, anti-mouse or anti-rabbit secondary antibodies (Jackson Immunoresearch) conjugated to Alexa 488, Alexa 594 or Alexa 633 (1:200), GFP-booster (Chromotek, 1:500), Alexa-594-conjugated Streptavidin (Jackson Immunoresearch, 1:100) and Alexa 568-conjugated phalloidin (Invitrogen, 1:500), were incubated for 1 hr at 37°C, followed by nuclear staining with DAPI (10 min, 300 ng/ml in PBS; Sigma). Fluorescence imaging was performed using an upright wide-field fluorescent microscope (Axioimager D1, Zeiss) or super-resolution microscopy (Leica SP8 Lightning and Zeiss LSM 880 Fast Airyscan) with 63x Plan ApoChromat NA1.4. For cilia measurements and counting, primary cilia from at least fifteen different fluorescent micrographs taken for each experimental condition were analyzed using the ruler tool from Adobe Photoshop. Cilia frequency was obtained dividing the number of total cilia by the number of nuclei on each micrograph. Number of cells per micrograph was similar in both TBS and control fibroblasts. To estimate the level of fluorescence in a determined region, we used the mean intensity obtained by ImageJ. To obtain the signal histograms on *Figure 2C–D*, we used the plot profile tool in FIJI.

## qRT-PCR analysis

Shh-LIGHT2 cells were starved for 48 hr. Total RNA was obtained with EZNA Total RNA Kit (Omega) and quantified by Nanodrop spectrophotometer. cDNAs were prepared using the SuperScript III First-Strand Synthesis System (Invitrogen) in 10 µl volume per reaction. *LUZP1, GAPDH, Gli1, Ptch1*, and *Rplp0* primers were tested for efficiency and products checked for correct size before being used in test samples. qPCR was done using PerfeCTa SYBR Green SuperMix Low (Quantabio). Reactions were performed in 10 µl, adding 1 µl of cDNA and 0.5 µl of each primer (10 µM), in a CFX96 thermocycler (BioRad) using the following protocol: 95°C for 10 min and 40 cycles of 95°C for 10 s and 55–60°C for 30 s. Melting curve analysis was performed for each pair of primers between 65°C and 95°C, with 0.5°C temperature increments every 5 s. Relative gene expression data were analyzed using the ΔΔCt method. Reactions were done in triplicates and results were derived from at least three independent experiments normalized to *GAPDH* and *Rplp0* and presented as relative expression levels. Primer sequences: *LUZP1-F*: 5′-GGAATCGGGTAGGAGACACCA-3′; *LUZP1-R*: 5′-TTCCCAGGCAGTTCAGACGGA-3; *GAPDH-F*: 5′-AGCCACATCGCTCAGACAC-3′; *GAPDH-R*: 5′-GCCCAATACGACCAAATCC-3′; *Gli1-F*: 5′-AGCCTTCAGCAATGCCAGTGAC-3′; *Gli1-R*: 5′-GTCAG-GACCATGCACTGTCTTG-3′; *Ptch1-F*: 5′-AAGCCGACTACATGCCAGAG-3′; *Ptch1-R*: 5′-TGATGCCATCTGCGTCTACCAG-3′, *Rplp0-F*: 5′-ACTGGTCTAGGACCCGAGAAG-3′; *Rplp0-R*: 5′-CTCCCACCTTGTCTCCAGTC-3′.

## Luciferase assays

Shh-LIGHT2 cells were starved for 48 hr, and treated or not for the last 24 hr with purmorphamine (5 µM, ChemCruz) to induce Shh signaling pathway. Firefly luciferase expression was measured using the Dual-Luciferase Reporter Assay System (Promega) according to the manufacturer's instructions. Luminescence was measured and data were normalized to the Renilla luciferase readout. For each construct, luciferase activity upon purmorphamine treatment was divided by the activity of cells before treatment to obtain the fold change value. Experiments were performed with both biological (n = 3) and technical (n = 6) replicates.

## Statistical analysis

Statistical analysis was performed using GraphPad 6.0 software. Data were analyzed by Shapiro-Wilk normality test and Levene´s test of variance. We used two-tailed unpaired Student´s t-test or Mann Whitney-U tests for comparing two groups, One-way ANOVA or Kruskall-Wallis and the corresponding post-hoc tests for more than two groups and two-way ANOVA to compare more than one variable in more than two groups. P values were represented by asterisks as follows: (*) p-value<0.05; (**) p-value<0.01; (***) p-value<0.001; (****) p-value<0.0001. Differences were considered significant when p<0.05. Values used for graphical representations and statistical analysis are available in Source Data 2.

## Acknowledgements

RB acknowledges A Cenigaonandia for her assistance in the experiments. We are grateful to the Fundación Inocente, Inocente for their support. We thank the Servicio General de Microscopía

Analítica y de Alta Resolución en Biomedicina, SGIker at the UPV/EHU. We also acknowledge funding by grants BFU2017-84653-P (MINECO/FEDER, EU), SEV-2016–0644 (Severo Ochoa Excellence Program), 765445-EU (UbiCODE Program), SAF2017-90900-REDT (UBIRed Program), IT634-13 (Basque Country Government) and POSTD19048BOZA (Fundación Científica AECC). Additional support was provided by the Department of Industry, Tourism, and Trade of the Basque Country Government (Elkartek Research Programs) and by the Innovation Technology Department of the Bizkaia County. FE is at Proteomics Platform, member of ProteoRed-ISCIII (PT13/0001/0027) and CIBERehd. AC acknowledges the Basque Department of education (IKERTALDE IT1106-16), the MCIU (SAF2016-79381-R (FEDER/EU)), the AECC (IDEAS175CARR; GCTRA18006CARR), La Caixa Foundation (HR17-00094) and the European Research Council (Starting Grant 336343, PoC 754627, Consolidator grant 819242). CIBERONC was co-funded with FEDER funds.

## Additional information

### Funding

| Funder | Grant reference number | Author |
|---|---|---|
| Ministerio de Economía y Competitividad | BFU2017-84653-P | Rosa Barrio |
| Ministerio de Economía y Competitividad | SEV-2016-0644 | Arkaitz Carracedo<br>Felix Elortza<br>James D Sutherland<br>Rosa Barrio |
| Ministerio de Economía y Competitividad | SAF2017-90900-REDT | Rosa Barrio |
| European Commission | 765445-EU | Orhi Barroso-Gomila<br>James D Sutherland<br>Rosa Barrio |
| Basque Government | IT634-13 | Arkaitz Carracedo |
| Asociacion Espanola Contra el Cancer | POSTD19048BOZA | Arkaitz Carracedo |
| Instituto de Salud Carlos III | PT13/0001/0027 | Arkaitz Carracedo |
| Basque Government | IKERTALDE IT1106-16 | Arkaitz Carracedo |
| Ministerio de Ciencia, Investigacion y Universidades | SAF2016-79381-R | Arkaitz Carracedo |
| Asociacion Espanola Contra el Cancer | IDEAS175CARR | Arkaitz Carracedo |
| Asociacion Espanola Contra el Cancer | GCTRA18006CARR | Arkaitz Carracedo |
| La Caixa Foundation | ID 100010434, agreement LCF/PR/HR17 | Arkaitz Carracedo |
| European Commission | 336343 | Arkaitz Carracedo |
| European Commission | PoC 754627 | Arkaitz Carracedo |
| European Commission | 819242 | Arkaitz Carracedo |
| Ministerio de Economía y Competitividad | RYC-2016-20480 | Olatz Pampliega |
| International Brain Research Organization | ReturnHomeFellowship18-3 | Olatz Pampliega |
| Ministerio de Ciencia e Innovación | RTI2018-097948-A-I00 | Olatz Pampliega |
| Instituto de Salud Carlos III | PT13/0001/0027 | Felix Elortza |
| Instituto de Salud Carlos III | CIBERehd | Felix Elortza |

The funders had no role in study design, data collection and interpretation, or the decision to submit the work for publication.

## Author contributions

Laura Bozal-Basterra, Conceptualization, Data curation, Formal analysis, Validation, Investigation, Methodology, Writing - original draft, Writing - review and editing; María Gonzalez-Santamarta, Investigation, Methodology; Veronica Muratore, Aitor Bermejo-Arteagabeitia, Carolina Da Fonseca, Orhi Barroso-Gomila, Mikel Azkargorta, Ibon Iloro, Olatz Pampliega, Ricardo Andrade, Natalia Martín-Martín, Arkaitz Carracedo, Felix Elortza, Methodology; Tess C Branon, Alice Y Ting, Jose A Rodríguez, Resources; James D Sutherland, Conceptualization, Resources, Supervision, Investigation, Methodology, Writing - original draft, Writing - review and editing; Rosa Barrio, Conceptualization, Formal analysis, Supervision, Funding acquisition, Writing - original draft, Project administration, Writing - review and editing

## Author ORCIDs

Mikel Azkargorta (iD) http://orcid.org/0000-0001-9115-3202
Olatz Pampliega (iD) http://orcid.org/0000-0002-7924-6374
Alice Y Ting (iD) http://orcid.org/0000-0002-8277-5226
Rosa Barrio (iD) https://orcid.org/0000-0002-9663-0669

## Ethics

Human subjects: The use of human samples in this study was approved by the institutional review board (Ethics Committee at CIC bioGUNE) and appropriate informed consent was obtained from human subjects or their parents. protocol P-CBG-CBBA-2111).

## Decision letter and Author response

Decision letter https://doi.org/10.7554/eLife.55957.sa1
Author response https://doi.org/10.7554/eLife.55957.sa2

# Additional files

## Supplementary files

- Source data 1. Identification of LUZP1 interactors by proximity proteomics.
- Source data 2. Values used for graphical representations and statistical analysis.
- Supplementary file 1. Key Resources Table.
- Transparent reporting form

## Data availability

All data generated or analysed during this study are included in the manuscript and supporting files.

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
