## [Decision Letter]

**Acceptance summary:**

This paper provides biochemical and cell biological evidence demonstrating that the leucine-zipper protein LUZP1 is a cytoskeletal regulator involved in controlling ciliogenesis and Sonic Hedgehog signaling, and that altered function of LUZP1 may contribute to the Townes-Brocks Syndrome.

**Decision letter after peer review:**

[Editors’ note: the authors submitted for reconsideration following the decision after peer review. What follows is the decision letter after the first round of review.]

Thank you for submitting your work entitled "LUZP1, a novel regulator of primary cilia and the actin cytoskeleton, is altered in Townes-Brocks Syndrome" for consideration by *eLife*. Your article has been reviewed by three peer reviewers, and the evaluation has been overseen by a Reviewing Editor and a Senior Editor. The reviewers have opted to remain anonymous.

Our decision has been reached after consultation between the reviewers. Based on these discussions and the individual reviews below, we regret to inform you that your work will not be considered further for publication in *eLife*.

While the reviewers found your manuscript interesting, they also raised several concerns implying that your manuscript would require very substantial revision to be acceptable for publication. One major concern is the quality of the LUZ1P centriole localization data, and the lack of several critical control experiments. In addition, there are some important concerns regarding the *Shh* signaling data and the proposed mechanistic link between LUZP1 and Townes-Brocks Syndrome. Although we are unable to consider your work for publication at this stage, we will be prepared to reconsider a substantially revised version of your manuscript, which would address the comments of the reviewers. If you decide to submit a revised manuscript, it will be treated as a new submission, but we will do our best to send it to the same reviewers.

Reviewer #1:

In their manuscript, "LUZP1, a novel regulator of primary cilia and the actin cytoskeleton, is altered in Townes-Brocks Syndrome", Bozal-Basterra et al. identify LUZP1 as a key component regulating cilia and actin dynamics and propose it may be key in the etiology of Townes-Brocks Syndrome (TBS). The authors propose that LUZP1 is central to the mechanism linking cytoskeletal rearrangements and cilia assembly, which in turns impacts cell signaling. Based on the interaction of LUZP1 with truncated SAL1, which causes TBS, the authors link LUZP1 to TWS. The strength of the manuscript is the extensive data which are generally well controlled and the data supporting a detailed mechanism of how LUZP1 functions normally at centrosomes to suppress ciliogenesis, stabilize F-actin and regulate *Shh* signaling. Additionally, the authors show LUZP1 is diminished in the presence of truncated SAL1 which in turn promotes ciliogenesis and abnormal *Shh* signaling. Thus, the authors present a solid cell biology story.

My enthusiasm for these data supporting the conclusion that this LUZP1 mechanism may underlie TBS is not high. It is one possibility but the mechanistic link is not secure. The data in the paper is in cell lines and it is not clear that the mechanism underlies the phenotypes in the patients. While the authors point out that Luzp1 mouse mutants display some overlapping phenotypes with TBS patients, the major reported phenotype in the mice involves neural tube closure defects and increased *Shh* expression which are not apparent in TBS. The authors show that there is increased *Shh* response in LUZP1 mutant cells which aligns with the authors' previous work in cell lines with truncated SAL1. However, it is not clear that increased *Shh* response would lead to TBS phenotypes as proposed. Generally how the proposed LUZP1 mechanism links to the tissue-specific patient phenotypes remains unclear.

Reviewer #2:

In this manuscript Bozal-Basterra et al. analyse the function of the leucine zipper protein LUZP1 in cilia formation. They identified LUZP1 as BioID interactor of SALL1. Mutations in SALL1 are associated with Townes-Brocks Syndrome (TBS1). In the first part, they show that LUZP1 associates with the proximal end of centrioles and interacts with CEP97 and CCP110 (although they are at the distal end of centrioles). In the second part of the manuscript, the authors show that LUZP1 associates with the actin cytoskeleton and that its deletion in NIH3T3 fibroblasts promotes cilia formation even without serum starvation. They continue with data suggesting that LUZP1 is subject to degradation by the proteasome. Furthermore, they suggest that SALL1 regulates LUZP1 stability. In particular, a truncation in SALL1 that arises in TBS1 seems to reduce LUZP1 levels. This may cause cilia formation in interphase and activation of *Shh*.

The middle part of the manuscript is the most significant, convincing and exciting one. It is convincingly shown that LUZP1 suppresses the formation of cilia in interphase via the actin cytoskeleton and its degradation in G2/M and G0 probably contributes to primary cilia formation. The first part of the manuscript is much less convincing and I am not sure what the centriole association of LUZP1 means. The impact of the final part of the paper, the regulation of LUZP1 by SALL1, is limited by the very mild effect of 293335 on LUZP1 levels (20%) (Figure 7C).

This is an interesting paper, however, it needs clearly more work to make it suitable for publication in *eLife*. I would remove the centriole localization of LUZP1 from the manuscript and invest more efforts in showing the mechanistic principals and relevance of the regulation of LUZP1 by SALL1 and the proteasome.

1) The images in Figure 2 are unusual. The authors use the Z stacks with low resolution to reconstitute the side views of centrioles. This results in distorted images. To increase quality of the images, the authors have to analyse centrioles in side views in X-Y. This will give them a much better localization of LUPZ1 relative to know proteins of the centriole/centrosome. Furthermore, to show specificity of the LUZP1 signal at centrosomes of RPR1 and U2OS cells, the authors have to deplete LUZP1 and show reduction or disappearance of the signal. Further suggest to use PCNT as a PCM marker in IF.

2) Figure 3A and B should have gamma-tubulin staining. However, according to the figure it shows acetylated alpha-tubulin. According to the legend it has acetylated alpha-tubulin and gamma-tubulin both in magenta. I do not agree with the statement: "Our results showed that LUZP1 was markedly decreased in TBS275 cells.… and LUZP1 was visualized as two rings that encircle each of the centrioles,.…". First, I do not see the decrease of LUZP1 and second the LUZP1 signal localizes dot-like in the periphery of centrioles. It is not a circle.

3) In Figure 3 that authors show that LUZP1 associates with CCP110 and CEP97 in pull downs and IP experiments. How does this fit with the localization of LUZP1 at the proximal end of centrioles?

4) I do not understand the data points in Figure 8B – how do the author explain this big variation?

5) Figure 8 shows transient transfection experiments without controlling expression levels.

6) Why is the loading control GAPDH not present in all lanes of Figure 7E?

Reviewer #3:

In this manuscript, Bozal-Basterra and colleagues identified the leucine-zipper containing protein LUZP1 as a novel component of the centrosome and actin cytoskeleton. They generated LUZP1 KO cells, which had higher rates of ciliogenesis and longer primary cilia, increased *Shh* signaling and decreased F-actin levels. Moreover, overexpression of LUZP1 repressed ciliogenesis and increased F-actin levels. In previous work, they identified LUZP1 as a proximity interactor of SALL1 protein, which is mutated in Townes-Brocks syndrome. Following on this finding, they identified physical and proximity interactions between truncated and full-length SALL1 and LUZP1 and showed that TBS cells had a reduction in LUZP1 and F-actin levels. Finally, they show that truncated SALL1 promotes LUZP1 degradation through regulation of its ubiquitination. Based on these findings, they propose LUZP1 as a key factor integrating cytoskeletal changes to cilia formation and function.

The manuscript is of general interest to the field because LUZP1 mutations were previously linked to cardiovascular defects and cranial NTC in mice and thus dissecting the function and regulation of this protein will contribute to our understanding of disease mechanisms. Additionally, the results of this paper identify LUZP1 as a negative regulator of ciliogenesis as a protein at the intersection of centrosome and actin cytoskeleton. However, given the complexity of interactions and functions associated with LUZP1 presented in this paper and published previously, it is not clear how LUZP1 mediates the reported functions, which weakens the model they present. Moreover, for some of the results, the data included is not sufficient to derive these conclusions and appropriate controls are missing. The following points must be addressed before publication of this paper in *eLife*:

1) One of the weakest points of the manuscript is the endogenous localization of the protein. The data presented for endogenous localization in Figure 2, Figure 3 and Figure 4 are somewhat contradictory. For example, in Figure 4, actin staining but not the centrosome staining is visible. However, in Figure 2, there is strong centrosome staining for LUZP1. Finally, in Figure 3, there are also LUZP1-positive puncta around the centrosome. To test the specificity, the authors must stain wt and LUZP1-/- cells with the antibody and include data on which cellular localizations of LUZP1 is specific. Given that their model is based on LUZP1 localization to both the centrosome and actin cytoskeleton, this point is very important.

2) The presented TurboID data presented for LUZP1 lacks the required controls and analysis. First, the localization and biotinylation activity of TurboID-LUZP1 must be shown by immunofluorescence and immunoblotting. Does the biotinylation activity reflect localization of endogenous protein at the centrosome and actin cytoskeleton? Second, what were the controls used for distinguishing the specific proximity interactors of LUZP1 form the non-specific for the application of the TurboID approach (there are none included in the paper)? An empty TurboID control must be included as a control.

3) Wang and Nakamura, 2019 paper identified LUZP1 as an interactor of Filamin, in which they mapped the actin binding site to 400-500 aa region of LUZP1. The authors can perform rescue experiments with this fragment in LUZP1-/- cells to distinguish the contribution of actin binding activity of LUZP1 to ciliogenesis from its centrosomal localization.

4) To corroborate the relationship between LUZP1, actin and ciliogenesis, the authors should test whether cytoD treatment antagonizes the effect of YFP-LUZP1 overexpression on ciliogenesis?

5) LUZP1 KO cells must be validated by immunofluorescence, immunoblotting and/or sequencing of the mutated region to show the frameshift.

6) The quality of the high resolution data in Figure 2 must be improved, the images look distorted in some, maybe due to deconvolution settings that were applied. Additionally, PCM1 staining in Figure 2E does not reflect the granular localization of centriolar satellites.

7) Given that LUZP1 and CP110/Cep97 localize to different parts of the centrosome (distal versus proximal), what is the authors hypothesis about how LUZP1 regulates CP110/Cep97 localization at the centrosome? Do these interactions occur at the cytoplasm? The model presented at the end of the paper focuses on LUZP1, SALL1, actin and ciliogenesis. How do CP110 and Cep97 fit to this model based on their data?

8) The authors switch between using RPE1 cells, U2OS cells, HEK293 cells, NIH3T3 cells and patient fibroblasts in different experiments. For the experiments related to LUZP1 phenotypic characterization, results for the same line should be included (additional cell lines can be kept as long as same one is carried along for all).

[Editors’ note: further revisions were suggested prior to acceptance, as described below.]

Thank you for submitting your article "LUZP1, a novel regulator of primary cilia and the actin cytoskeleton, is a contributing factor in Townes-Brocks Syndrome" for consideration by *eLife*. Your article has been reviewed by three peer reviewers, and the evaluation has been overseen by a Reviewing Editor and Anna Akhmanova as the Senior Editor. The following individual involved in review of your submission has agreed to reveal their identity: Elmar Schiebel (Reviewer #1).

The reviewers have discussed the reviews with one another and the Reviewing Editor has drafted this decision to help you prepare a revised submission.

Summary:

In this revised manuscript Bozal-Basterra et al. report on the leucin-zipper protein LUZP1 as an interactor of a truncated form SALL1 that is causing in a dominant way Townes-Brocks-Syndrome. They provide evidence that LUZP1 associates with cilia and actin filaments and that loss of LUZP1 reduces F-actin levels and impacts on cilia function. Interestingly, the truncated SALL1 increases LUZP1 ubiquitination and degradation, and although the underlying mechanism is unclear this suggests that reduced levels of LUZP1 might contribute to how the truncated SALL1 causes the disease. The manuscript is generally convincing and of importance even though the authors do not demonstrate a causative link of reduced LUZP1 levels to TBS.

Essential revisions:

The authors propose that LUZP1 is central to the mechanism linking cytoskeletal rearrangements and cilia assembly, which in turn impacts cell signaling and suggest that this underlies the etiology of TBS. While this speculation is certainly possible, the authors do not demonstrate a causative link of this mechanism to TBS. Additionally, the mechanism through which SALL1 and the proteasome regulate LUZP1 is not fully developed. In order for this manuscript to be acceptable for publication, the authors must modify the manuscript to very clearly indicate that they have not nailed the etiology of the disease. Furthermore, all suggestions about the therapeutic implications of their work should also be removed.

---

## [Author Response]

[Editors’ note: the authors resubmitted a revised version of the paper for consideration. What follows is the authors’ response to the first round of review.]

While the reviewers found your manuscript interesting, they also raised several concerns implying that your manuscript would require very substantial revision to be acceptable for publication. One major concern is the quality of the LUZ1P centriole localization data, and the lack of several critical control experiments. In addition, there are some important concerns regarding the Shh signaling data and the proposed mechanistic link between LUZP1 and Townes-Brocks Syndrome. Although we are unable to consider your work for publication at this stage, we will be prepared to reconsider a substantially revised version of your manuscript, which would address the comments of the reviewers. If you decide to submit a revised manuscript, it will be treated as a new submission, but we will do our best to send it to the same reviewers.Reviewer #1:In their manuscript, "LUZP1, a novel regulator of primary cilia and the actin cytoskeleton, is altered in Townes-Brocks Syndrome", Bozal-Basterra et al. identify LUZP1 as a key component regulating cilia and actin dynamics and propose it may be key in the etiology of Townes-Brocks Syndrome (TBS). The authors propose that LUZP1 is central to the mechanism linking cytoskeletal rearrangements and cilia assembly, which in turns impacts cell signaling. Based on the interaction of LUZP1 with truncated SAL1, which causes TBS, the authors link LUZP1 to TWS. The strength of the manuscript is the extensive data which are generally well controlled and the data supporting a detailed mechanism of how LUZP1 functions normally at centrosomes to suppress ciliogenesis, stabilize F-actin and regulate Shh signaling. Additionally, the authors show LUZP1 is diminished in the presence of truncated SAL1 which in turn promotes ciliogenesis and abnormal Shh signaling. Thus, the authors present a solid cell biology story.My enthusiasm for these data supporting the conclusion that this LUZP1 mechanism may underlie TBS is not high. It is one possibility but the mechanistic link is not secure. The data in the paper is in cell lines and it is not clear that the mechanism underlies the phenotypes in the patients.

We appreciate the positive comments of the reviewer.

While we performed numerous experiments in established, widely-used cell lines in order to characterize LUZP1, including a novel CRISPR/Cas9-mediated LUZP1 knockout model (in NIH3T3 fibroblasts), the experiments linking LUZP1 to TBS were performed in primary patient-derived TBS^275^ and control dermal fibroblasts (featured in Figures 3A-C, 4A-C, 9A-B, 10A-C). Access to other cell types is impractical, even more because TBS is a very rare genetic syndrome. We attempted to obtain LUZP1-KO primary cells and/or mice from the Chang Lab (Hsu et al., 2008) but we were unsuccessful. Primary cilia analysis in that model would have been informative. In the future, we aim to analyze LUZP1 levels and cilia dysfunction in a mouse TBS model (Kiefer et al., 2003), even if the mouse has variable penetrance and phenocopies some (but not all) hallmarks of human TBS.

Caveats aside, throughout the manuscript, we present evidence that suggest a relationship between LUZP1 and TBS:

– Interaction of LUZP1 with truncated SALL1 by BioID, pulldown and immunoprecipitation (Figure 1 and Figure 1—figure supplement 2A)

– Truncated SALL1 can cause reduction of LUZP1 levels through ubiquitination and proteasomal degradation (WB: Figures 1A, 3D, Figure 9C-F), and in TBS cells (WB: Figure

9A-B; IF: Figures 3B, 4A, 10C)

– Cellular loss of LUZP1 promotes more and longer cilia, phenocopying TBS cells (Figure

6A, B, C)

– Loss of LUZP1 leads to a reduction of CCP110 levels in the mother centriole, phenocopying TBS cells (Figure 6D, E)

– Loss of LUZP1 leads to alterations in SHH pathway similar to TBS (Figure 7F-H)

– Importantly, cytoskeletal and cilia phenotypes in TBS cells are modified by exogenous

LUZP1 (Figure 10)

While the evidence for a LUZP1-TBS link is supportive, we believe that LUZP1 is only one of multiple factors affected by the dominant action of truncated SALL1, which collectively contribute to the TBS phenotype. We clarified this repeatedly in the text and state that LUZP1 might be a “contributing factor” to TBS.

While the authors point out that Luzp1 mouse mutants display some overlapping phenotypes with TBS patients, the major reported phenotype in the mice involves neural tube closure defects and increased Shh expression which are not apparent in TBS.

Mutant mice for combinations of SALL gene (*Sall1*, *Sall2*, *Sall4*) display exencephaly (Böhm et al., 2008) and a TBS mouse model displays exencephaly when in homozygosity (Kiefer et al., 2003). These defects are the same as the exencephaly reported for LUZP1 KOs. In *Xenopus*, *Sall1* morphants also present neural tube defects (Exner et al., 2017). This information has been now added to the Discussion in a specific section entitled “The role of LUZP1 in TBS phenotype”. To our knowledge, *Shh* expression status has not been examined in these mouse/*Xenopus* models, in the Rauchman TBS mouse model, or in any human TBS-derived samples. Likewise, the septal defects in cardiac development reported for the *Luzp1* KO mice are similar to septal defects reported in up to 25% of TBS patients

(https://www.ncbi.nlm.nih.gov/pubmed/11102974/, https://www.ncbi.nlm.nih.gov/books/NBK1445/)

In the case of *Luzp1* KO mice, it would be informative to check cilia length and frequency, as well as response to *Shh*, to see whether it coincides with our LUZP1-KO or TBS cellular models. Other TBS phenotypes, such as subtle morphological changes to digits, small kidney cysts, or hearing impairment, may have been overlooked in the *Luzp1* KO mice. As mentioned, we could not obtain *Luzp1* KO mice or derived MEFs. Since the authors have retired/relocated, it is unclear if any mice/samples/antibodies from their report have been maintained.

The authors show that there is increased Shh response in LUZP1 mutant cells which aligns with the authors' previous work in cell lines with truncated SAL1. However, it is not clear that increased Shh response would lead to TBS phenotypes as proposed.

Whether TBS developmental malformations all arise from higher basal activity or increased expression of *Shh* targets as we see in our cell models is difficult to discern. The characteristic TBS phenotypes include digit malformations, misshapen outer ears and hearing problems, and gastrointestinal anomalies, which can all be linked with misregulated *Shh* signaling. Since primary cilia can also influence other developmental signaling pathways (Wnt, TGFbeta, Notch, FGF, PDGF), ciliary changes driven by truncated SALL1 could have effects beyond *Shh*.

We added this information now in the Discussion section in a specific section entitled “The role of LUZP1 in TBS phenotype”.

Preaxial polydactyly has been associated with ectopic SHH expression in limbs. Examples:

Johnson et al., 2014; Lettice et al., 2003, 2008; Dunn et al., 2011.

Ectopic activation of *Shh* signaling is seen in *Kif7* mouse models, characterized by polydactyly: Zhulyn and Hui, 2015; Ibisler A et al. Mol Syndromol. 2015. PMID: 26648833; Lam et al. J Hand Surg Eur Vol. 2019. PMID: 29587601; Amano et al. G3 (Bethesda). 2017. PMID: 28710291; Liem et al., 2009.

Misregulation of the *Shh* pathway has been also related to anal stenosis/imperforate anus. For example, overexpression of *Shh* can induce the expression of *Bmp4* and *Hoxd13* in hindgut mesoderm (Roberts et al., 1995 *Gli2/Gli3* loss-of-function is related to anal malformations: “*Gli2*-/- mice exhibit imperforate anus, whereas Gli3-/- mice display less severe anal stenosis (Mo et al., 2001), both of which are observed in TBS. In humans, some individuals with Pallister-Hall syndrome [PHS (MIM: 165240)], which is caused by mutations in the *Shh* effector *GLI3*, also exhibit anorectal malformations, such as imperforate anus (Kang et al., 1997).

*Shh* has also been linked to deafness and dysplastic ears. For example, *Shh* signaling regulates sensory cell formation and auditory function in mice and humans (Driver et al., 2008), as well inner ear patterning (Bok et al. Development, 2007. PMID 17395647). The external ear is thought to derive from the first and second pharyngeal pouches, the correct patterning of which is dependent on *Shh* signaling (MooreScott BA, Manley NR. Dev. Biol. 2005. PMID 15680353).

TBS show similar phenotypes to those of VACTERL association (acronym for vertebral anomalies, anal atresia, congenital cardiac disease, tracheoesophageal fistula, renal anomalies, radial dysplasia, and other limb defects). Due to the similarity of phenotypes of *Gli* mutant mice, it has been proposed that defective *Shh* signaling leads to a spectrum of developmental anomalies in mice strikingly similar to those of VACTERL (Kim et al. Clin. Genet, 2001. PMID: 11359461; Lubinsky Am J Med Genet A. 2015. PMID: 26198446; Lubinsky Am J Med Genet A. 2015. PMID: 26174174).

Therefore, based on the previous literature and the fact that TBS patient-derived fibroblasts show alterations in *Shh* signaling (Bozal-Basterra et al., 2018), we believe that an association between TBS phenotypes and alterations of *Shh* pathway during development is plausible. However, to clarify for readers, we state in the manuscript that changes in *Shh* signaling have not been checked in other TBS patient-derived tissues (e.g. embryos), nor in mouse models for TBS.

Generally how the proposed LUZP1 mechanism links to the tissue-specific patient phenotypes remains unclear.

The effect of LUZP1 on the actin cytoskeleton in general, or especially as it pertains to centrosome/cilia function, could definitely contribute to TBS phenotypes. Our conclusion that LUZP1 and TBS phenotypes are linked is based on observations in cells and comparisons between reports on LUZP1-KO mouse, TBS models/case reports, and other ciliopathies/cilia-related phenotypes:

– Both TBS and *Luzp1* KO cells show longer and more frequent cilia than control cells, which correlates with a lower occupancy of the mother centriole by CCP110. This is accompanied by an increase in *Shh* response (Bozal-Basterra et al., 2018; and this work).

– Longer cilia have been related previously to polydactyly (Liem et al., 2009; He et al. Nat Cell Biol. 2014. PMID: 24952464).

– TBS patient-derived fibroblasts show lower levels of LUZP1 than control cells. Increasing the levels of LUZP1 in TBS cells reduces the number of ciliated cells. –

Regarding the tissue-specific phenotypes:

It is important to bear in mind that SALL1 is a highly regulated factor that is not expressed in all tissues. For instance, mRNA has been detected in human brain, liver and kidneys (Kohlhase et al. Genomics. 1996 Dec 15;38(3):291-8. PMID: 8975705). In mice, SALL1 protein expression is very prominent in brain and limbs, as well as kidneys, lens, olfactory bulbs, heart, primitive streak and the genital tubercle. These correspond to the organs affected in human TBS (Buck et al. Mech Dev. 2001. PMID: 11404093).

LUZP1 expression appears to be broader. Using an in-house antibody (that we were unable to obtain) and western blotting (Hsu et al., 2008), LUZP1 was reported to be primarily expressed in brain, with weak expression in heart, lungs, and kidneys. More recent validation by the Human Protein Atlas, using two independent rabbit polyclonal antibodies (one of which we have used in our report, Σ HPA028506), shows a more extensive distribution. 45 different adult human tissues showed LUZP1 expression with the two antibodies, with 36 tissues also showing correlation with RNA levels (https://www.proteinatlas.org/ENSG00000169641-

LUZP1/tissue). These antibodies have not been used thus far to examine LUZP1 distribution and levels in developing mouse embryos.

We expect that the tissues affected are those that express the truncated SALL1 and LUZP1, and undergo morphological changes dependent on actin cytoskeleton and/or cilia-based signaling. This argues for a partial, but not necessarily complete, overlap in TBS and LUZP1-KO phenotypes and is compatible with a role for LUZP1 in TBS etiology. Also, we believe that LUZP1 is one of multiple factors affected by the dominant action of truncated SALL1, so only a partial overlap is expected. To make this more evident to the readers, we included a more extended explanation in the Discussion section.

Reviewer #2:In this manuscript Bozal-Basterra et al. analyse the function of the leucine zipper protein LUZP1 in cilia formation. They identified LUZP1 as BioID interactor of SALL1. Mutations in SALL1 are associated with Townes-Brocks Syndrome (TBS1). In the first part, they show that LUZP1 associates with the proximal end of centrioles and interacts with CEP97 and CCP110 (although they are at the distal end of centrioles). In the second part of the manuscript, the authors show that LUZP1 associates with the actin cytoskeleton and that its deletion in NIH3T3 fibroblasts promotes cilia formation even without serum starvation. They continue with data suggesting that LUZP1 is subject to degradation by the proteasome. Furthermore, they suggest that SALL1 regulates LUZP1 stability. In particular, a truncation in SALL1 that arises in TBS1 seems to reduce LUZP1 levels. This may cause cilia formation in interphase and activation of Shh.The middle part of the manuscript is the most significant, convincing and exciting one. It is convincingly shown that LUZP1 suppresses the formation of cilia in interphase via the actin cytoskeleton and its degradation in G2/M and G0 probably contributes to primary cilia formation. The first part of the manuscript is much less convincing and I am not sure what the centriole association of LUZP1 means. The impact of the final part of the paper, the regulation of LUZP1 by SALL1, is limited by the very mild effect of 293335 on LUZP1 levels (20%) (Figure 7C).

We appreciate the reviewer’s positive comments. While LUZP1 localizes both to centrioles and to actin filaments, we cannot say that its role in cilia regulation is through one structure or the other, or both. After our submission to bioRxiv and *eLife*, another LUZP1 preprint was submitted to bioRxiv which supports our conclusions, specifically that LUZP1 has a role in actin cytoskeleton and cilia formation (Goncalves et al. bioRxiv. https://doi.org/10.1101/736389). Also, in agreement with our results, the authors showed that exogenous LUZP1 could counteract the permissive role of CytoD on cilia formation.

While the reviewer mentions the mild variation of LUZP1 levels, the differences with control cells are statistically significant not only in 293^335^ cells, but also in 293 cells expressing exogenous truncated SALL1 (WB: Figures 1A, 3D, Figure 9C-F) and in TBS patient-derived cells (WB: Figure 9A-B; IF: Figures 3B, 4A-C, 10C ). Although the change is modest, it is consistent in the different models, therefore it might be biologically relevant for the TBS phenotype.

This is an interesting paper, however, it needs clearly more work to make it suitable for publication in eLife. I would remove the centriole localization of LUZP1 from the manuscript and invest more efforts in showing the mechanistic principals and relevance of the regulation of LUZP1 by SALL1 and the proteasome.

In order to understand how truncated SALL1 might be compromising LUZP1 function, it is only correct to consider all the subcellular sites where the protein is detected (actin filaments, centrosome, and the midbody). These sites are all labelled with LUZP1 antibody and with exogenous LUZP1-YFP (in WT and rescued LUZP1-KO cells). Therefore, we believe that it is important to keep manuscript data showing centriole localization of LUZP1 to allow us and others to pursue further work on the mechanisms at play in these diverse (but interrelated) subcellular locations.

We agree with the reviewer that the regulation of LUZP1 degradation by truncated SALL1 is an important subject to be investigated. From the literature and especially our BioID experiments, we have some clues for potential deubiquitinases (USP9X, USP14) and E3 ligases (*MIB1*) that might be involved in this regulation, but to screen candidates thoroughly and have a clear idea of this mechanism will require additional resources and time that does not justify delaying our report. In this paper, we aimed to characterize LUZP1 as a contributor to TBS phenotypes, with focus on the primary cilia. We expect that other factors affected by truncated SALL1, including those that regulate LUZP1/CP110 stability, will also have features that phenocopy TBS and will lead to interesting complementary studies in the future.

1) The images in Figure 2 are unusual. The authors use the Z stacks with low resolution to reconstitute the side views of centrioles. This results in distorted images. To increase quality of the images, the authors have to analyse centrioles in side views in X-Y. This will give them a much better localization of LUPZ1 relative to know proteins of the centriole/centrosome.

As suggested by the reviewer, we substituted the images in Figure 2 by 2D images. We took high resolution images and chose side views of the centrioles, using diverse markers.

Furthermore, to show specificity of the LUZP1 signal at centrosomes of RPR1 and U2OS cells, the authors have to deplete LUZP1 and show reduction or disappearance of the signal. Further suggest to use PCNT as a PCM marker in IF.

An immunofluorescence control for LUZP1 antibody specificity was included in the “old” Figure 5—figure supplement 1. We show this data now in the new main Figure 5 to make it more visible for the readers. Immunofluorescence of *Luzp1* CRISPR KO cells reveals no LUZP1 staining at centrosomes, nor at the cytoskeleton. LUZP1 antibody specificity is also shown by Western blot in Figure 5C.

Following the useful suggestion, we purchased PCNT mouse monoclonal antibody (Abcam ab28144) and performed new stainings and high resolution imaging using PCNT to label pericentriolar material, together with rat anti-centrin2 (centrioles) and rabbit anti-LUZP1. These images are featured in the new Figure 2C.

2) Figure 3A and B should have gamma-tubulin staining. However, according to the figure it shows acetylated alpha-tubulin. According to the legend it has acetylated alpha-tubulin and gamma-tubulin both in magenta. I do not agree with the statement: "Our results showed that LUZP1 was markedly decreased in TBS275 cells.… and LUZP1 was visualized as two rings that encircle each of the centrioles,.…". First, I do not see the decrease of LUZP1 and second the LUZP1 signal localizes dot-like in the periphery of centrioles. It is not a circle.

In the “old” Figure 3A/B, we simultaneously used mouse anti-acetylated alpha-tubulin to label ciliary axonemes, and mouse anti-gamma-tubulin to label basal bodies, or centrosomes in non-ciliated cells, together with the same anti-mouse fluorophore. This was done so we could use the remaining two channels for other markers (anti-rabbit, anti-rat primary Abs). These panels have now been changed in the new Figure 3 where we used mouse antiacetylated alpha-tubulin to label the primary cilia and rat centrin to label the centrioles in different channels.

On closer inspection, we agree with the reviewer that the reduction of LUZP1 at the centrosomes in TBS cells was not clearly evident in those images (“old” Figure 3A/B), taken on the confocal microscope and processed by reconstruction and Lightning software algorithm (Leica). Therefore we re-examined the labelled cells using wide-field fluorescence microscope, capturing multiple fields, using identical settings between the control and TBS samples. These images have been included in the new Figure 3A-C. Quantification of LUZP1 fluorescence at the centrioles (defined by CETN2 staining) shows a significant reduction of LUZP1 intensity in TBS cells compared to control, and this is even more reduced by starvation.

To avoid misunderstandings, we changed the text to say now that LUZP1 surrounds the centrioles at the proximal end of both centrioles.

3) In Figure 3 that authors show that LUZP1 associates with CCP110 and CEP97 in pull downs and IP experiments. How does this fit with the localization of LUZP1 at the proximal end of centrioles?

Our pulldown and coimmunoprecipitation experiments show interaction between LUZP1 and CCP110/CEP97. Also this interaction has been corroborated by our new proximity proteomics data. Nevertheless, these techniques (pulldown, co-immunoprecipitation, or proximity TurboID) do not always define direct interactions of the analyzed proteins. We specify this point in the new version of the manuscript to make it more clear to the readers.

We speculate that, even though localization by IF implies that LUZP1 is proximal and CCP110/CEP97 is distal on the centrioles, the centrosomal “environment” is in flux and dynamic. These proteins may encounter each other as steady-state patterns are established. Also, the proteins could be transported to the centrosome in complexes, which then disassemble and resolve into the patterns that we see by IF. LUZP1 and CCP110, as well as many other centrosomal proteins, are coiled-coil proteins that might be forming higher order assemblies, so the interaction could be through bridging factors.

4) I do not understand the data points in Figure 8B – how do the author explain this big variation?

Regarding “old” Figure 8B, now “new Figure 10B: As we mentioned in Materials and methods, primary cilia from at least fifteen different fluorescent micrographs taken for each experimental condition were analyzed using the ruler tool from Adobe Photoshop. Cilia frequency was obtained dividing the number of total cilia by the number of nuclei on each micrograph. In order to measure the cilia properly, we used 63x objective, which covers 2-4 cells per micrograph. Therefore, the division of cilia per total nuclei gives this type of variation.

5) Figure 8 shows transient transfection experiments without controlling expression levels.

We thank the reviewer for this comment, as it revealed a mistake in the previous submission. The experiments shown in “old” Figure 8 (now Figure 10) were not transient transfections but lentiviral transductions. As shown in Figure 5C, the levels of transduced LUZP1-YFP (at least in the rescued mouse 3T3 *Luzp1* KO cells) are not very different from the endogenous LUZP1 levels. Lentiviral transductions into primary human fibroblasts were done for immunofluorescence only. Since the same viral preparations were used for 3T3 and human fibroblast experiments, exogenous levels of LUZP1-YFP are expected to be similar to endogenous LUZP1.

6) Why is the loading control GAPDH not present in all lanes of Figure 7E?

We also thank the reviewer for this comment, as it revealed a mistake in the previous submission: GFP and GAPDH labels were mislabeled in old Figure 7E (now Figure 9E), while they were correct in the supplementary figure showing the full blots. This mistake has been corrected, with these blots included in new Figure 9E and Figure 9—figure supplement 1.

Reviewer #3:[…] The manuscript is of general interest to the field because LUZP1 mutations were previously linked to cardiovascular defects and cranial NTC in mice and thus dissecting the function and regulation of this protein will contribute to our understanding of disease mechanisms. Additionally, the results of this paper identify LUZP1 as a negative regulator of ciliogenesis as a protein at the intersection of centrosome and actin cytoskeleton. However, given the complexity of interactions and functions associated with LUZP1 presented in this paper and published previously, it is not clear how LUZP1 mediates the reported functions, which weakens the model they present. Moreover, for some of the results, the data included is not sufficient to derive these conclusions and appropriate controls are missing. The following points must be addressed before publication of this paper in eLife:1) One of the weakest points of the manuscript is the endogenous localization of the protein. The data presented for endogenous localization in Figure 2, Figure 3 and Figure 4 are somewhat contradictory. For example, in Figure 4, actin staining but not the centrosome staining is visible. However, in Figure 2, there is strong centrosome staining for LUZP1. Finally, in Figure 3, there are also LUZP1-positive puncta around the centrosome.

We believe that these discrepancies can be mostly attributed to the different microscopic techniques used, but we will attempt to clarify:

We describe three main sites of LUZP1 localization: actin filaments, centrosome/basal body, and the cytokinetic midbody. Actin filaments can be found throughout the cell, but most prominently in actin stress fibers that stretch across the basal side of cells. Centrosomes tend to stay close to the nucleus, usually in an apical manner, even more when the centrosome engages with the apical plasma membrane, converting to a basal body as the mother centriole serves to template the growing ciliary axoneme. The midbody is only seen briefly during cytokinesis and abscission, so very few cells in an unsynchronized population are at this stage.

Depending on the cell type, the distance between the basal and apical sides can vary, and confocal microscopy usually focuses at one level or the other to reveal the LUZP1 localization. Opening the pinhole (to allow more out-of-focus light and increasing plane thickness) or doing Z-stacks and reconstructions can reveal localizations to different planes in the same image. Some images were taken using wide-field fluorescence microscopy and often LUZP1 localization to the centrosome and actin stress fibers can be seen simultaneously.

While various imaging techniques and planes are shown, the LUZP1 localization at actin filaments, centrosomes and midbody is consistent in all the cell types that we have examined using specific antibodies, as well as with exogenous expression of LUZP1-YFP. After our submission to *eLife* and posting on bioRxiv, another preprint on LUZP1 was posted that strongly supports our conclusions (Goncalves et al. bioRxiv. https://doi.org/10.1101/736389). Using a different antibody, the authors also observed LUZP1 in actin filaments, centrosomes, and the midbody. Another recent study (Wang and Nakamura, 2019) also localizes LUZP1 to actin filaments.

Regarding LUZP1-positive puncta around the centrosome: In most of the stainings, LUZP1 intensity at the centrosome is much higher, but it is true that weaker punctae are often observed. We believe that the Lightning software algorithm (Leica) used in old Figure 3 increases the intensity of the puncta, which looked then more prominent than in other pictures. Old Figure 3A/B have been now substituted by fluorescence micrographs, which reflect better the reduction of intensity of LUZP1 at the centrosome in TBS cells.

We also note that a recent publication from the Pelletier and Raught groups used BioID to explore the proximal interactors of known centriolar satellite proteins (e.g. PCM1, CEP131; Gheiratmand et al., 2019). With multiple baits, they identified LUZP1, suggesting that these punctae mentioned by the reviewer could be sporadic centriolar satellites. In the LUZP1-TurboID experiments that we report here, as well as BioIDLUZP1 experiments presented in the Pelletier group preprint (posted on bioRxiv after ours), PCM1 is found as a proximal interactor to LUZP1. However, unlike PCM1, we have never seen strong, frequent, clustered LUZP1 punctae around the centrosome using antibodies or LUZP1-YFP fusion. As many centriolar satellite proteins eventually contact the centrosome, the proximity interactions with LUZP1 are likely occurring there, rather than in the satellites.

To test the specificity, the authors must stain wt and LUZP1-/- cells with the antibody and include data on which cellular localizations of LUZP1 is specific. Given that their model is based on LUZP1 localization to both the centrosome and actin cytoskeleton, this point is very important.

We agree with the reviewer that the proper control of the antibodies is very important. A control of the antibodies used for immunofluorescence was already included in the “old Figure 5—figure supplement 1”. We show this data now in the new Figure 5 to make it more visible for the readers. Immunofluorescence of *Luzp1* KO cells show no LUZP1 staining at actin filaments or centrosomes. The midbody is also LUZP1-negative in those cells (data not shown). The specificity of the antibodies is also shown by western blot in Figure 5C.

2) The presented TurboID data presented for LUZP1 lacks the required controls and analysis. First, the localization and biotinylation activity of TurboID-LUZP1 must be shown by immunofluorescence and immunoblotting. Does the biotinylation activity reflect localization of endogenous protein at the centrosome and actin cytoskeleton?

According to the reviewer’s request, we redid the TurboID-LUZP1 experiment in RPE1 cells, together with a matching control, and present here new proteomics data. GO analysis of the proteins enriched in TurboID-LUZP1 versus TurboID alone show similar enrichments as our initial experiment, with centrosomal and actin cytoskeletal networks prominently featured (new Figure 1 and Figure 1—figure supplement 2).

In new Figure 1—figure supplement 2B, we show that TurboID-LUZP1 expression levels are even lower than those of endogenous protein, so massive exogenous expression is not an issue. In addition, biotinylation activity detected by fluorescent streptavidin reflects the localization of the endogenous protein in centrosome and actin filaments by using Pericentrin and Phalloidin as markers (new Figure 1—figure supplement 2C).

Of note: No streptavidin was detected in microtubules. Author response image 1 shows RPE1 cells expressing TurboID-LUZP1 (labelled with BirA antibody), localized to the centrosome and out-of-focus actin filaments. The resulting biotinylation is visualized by fluorescent streptavidin, with centrosome and actin filament labelling. Dot-like biotinylation could reflect sporadic centriolar satellites as mentioned above.

**Author response image 1. sa2fig1:** 

Biotinylated proximal proteins were captured on streptavidin-conjugated agarose, eluted, separated by PAGE and visualized with Sypro-Ruby (Biorad) in Author response image 2. We considered that it is not necessary to add this figure to the manuscript, although we can add it upon the reviewer’s request.

Second, what were the controls used for distinguishing the specific proximity interactors of LUZP1 form the non-specific for the application of the TurboID approach (there are none included in the paper)? An empty TurboID control must be included as a control.

The best control for the BioID and TurboID experiments is always a matter of discussion. BirA and its derivatives are bacterial in origin and do not have a specific subcellular localization in mammalian cells, therefore localize throughout the cell. Abundant proteins (like actin) might be randomly biotinylated. Consequently, if this “control” protein set is subtracted from a localized BioID fusion (i.e. TurboID-LUZP1) the certain interactors might be overlooked, so enrichment analysis must be considered. Following the recommendations of the reviewer, we performed a new proximity proteomics analysis comparing three replicas of TurboID alone versus three replicas of TurboID-LUZP1, which is shown in the new Figure 1 and Figure 1—figure supplement 2 and Source data 1. A good overlap with our initial TurboID-LUZP1 dataset was observed.

3) Wang and Nakamura, 2019 paper identified LUZP1 as an interactor of Filamin, in which they mapped the actin binding site to 400-500 aa region of LUZP1. The authors can perform rescue experiments with this fragment in LUZP1-/- cells to distinguish the contribution of actin binding activity of LUZP1 to ciliogenesis from its centrosomal localization.

Wang and Nakamura, 2019, showed that in cells, tagged fimbacin 1-500 localizes to actin filaments, whereas fimbacin 1-400 does not. Recombinant fimbacin 400-500 can bind actin in a pulldown assay, but cannot bundle actin filaments. Also tagged fimbacin 360-729 does not fully colocalize with F-actin, suggesting that fimbacin 400500 may be necessary (but not sufficient) for proper localization. No mention was made of centrosomal localization. Therefore, we believe rescue experiments using that fragment would be inconclusive. Using LUZP1 knockout cells may help (to remove the complicating effects of endogenous LUZP1 and formation of multimeric forms), but it is unknown whether LUZP1 can form multimers with the closely-related FILIP1/FILIP1L (also filamin-interacting proteins with coiled-coil domains). In fact, FILIP1L was found as a proximal interactor of TurboID-LUZP1 (this work), as well as a proximal interactor of many centriolar satellite proteins (Gheiratmand et al., 2019).

In line with the reviewer’s suggestion, we created various LUZP1 constructs, but we could not find a construct that directed the localization of LUZP1 to centrosome versus actin filaments. And perhaps LUZP1 at the centrosome is also actin-associated, since actin filaments have been observed emanating from isolated centrosomes (Farina et al. Nat Cell Biol 2016; PMID 2665583). These mapping experiments will require a more detailed analysis, with biochemistry and possibly in vitro reconstitution, beyond the scope of the current work.

4) To corroborate the relationship between LUZP1, actin and ciliogenesis, the authors should test whether cytoD treatment antagonizes the effect of YFP-LUZP1 overexpression on ciliogenesis?

Following the reviewer’s suggestion, we performed CytoD treatment experiments. As expected, the treatment with CytoD increased significantly the ciliogenesis in RPE1 cells. Our data showed that the overexpression of LUZP1-YFP suppressed the positive effects of CytoD. The new CytoD results have been added to the manuscript in the new Figure 10—figure supplement 1. These results are in agreement with the results shown by Goncalves et al. (bioRxiv, https://doi.org/10.1101/736389).

Exogenous LUZP1 increases intensity of actin fibers in WT and TBS cells, as we have shown in Figure 10C. With the understanding that LUZP1 is regulator of actin crosslinking (i.e. bundling of actin filaments Wang and Nakamura, 2019), exogenous expression of LUZP1 may act to bundle residual fibers after CytoD treatment, or protect filaments from CytoD action. The exact roles of F-actin in ciliogenesis are debated (Copeland, 2019), so it is difficult to say where exactly CytoD-mediated inhibition of actin polymerization is having its effect (stress fibers, cortical actin, centrosome-associated actin), but exogenous LUZP1 does counteract it.

5) LUZP1 KO cells must be validated by immunofluorescence, immunoblotting and/or sequencing of the mutated region to show the frameshift.

The *Luzp1* KO cells were validated by immunofluorescence and Western blot. Those results constituted the “old Figure 5—figure supplement 1” and now are shown as main Figure 5.

We did sequence the DNA of CRISPR *Luzp1*^-/-^ cells. Two sgRNAs were used in the experiment. Using flanking primers for genomic PCR, amplicons were sequenced and we found that cells have a homozygous deletion of the sequence:

5’ccacctgcgatttaagttacagagcctgagccgccgcctcgatgagttagaggaagctacaaaaaacctccagagagcagagga tgagctcctggacctccaggacaaggtgatccaggcagagggcagcgactccagcacgctggctgagatcgaggtgctgcgccagc gg-3’.

This generates a frameshift and early stop codon, resulting in a short N-terminal peptide: MAELTNYKDAASNRY*. This information is now included in the Materials and methods section.

6) The quality of the high resolution data in Figure 2 must be improved, the images look distorted in some, maybe due to deconvolution settings that were applied. Additionally, PCM1 staining in Figure 2E does not reflect the granular localization of centriolar satellites.

As suggested by reviewers 2 and 3, we substituted Figure 2 with new 2D images. We took high resolution images and chose side views of the centrioles, using diverse markers.

In relation to the PCM1 staining, we substituted the panel for another image in which the centriolar satellites were more visible.

7) Given that LUZP1 and CP110/Cep97 localize to different parts of the centrosome (distal versus proximal), what is the authors hypothesis about how LUZP1 regulates CP110/Cep97 localization at the centrosome? Do these interactions occur at the cytoplasm?

Our pulldown and coimmunoprecipitation experiments show interaction between LUZP1 and CCP110/CEP97. Also this interaction has been corroborated by our new proximity proteomics data. Nevertheless, these techniques (pulldown, co-immunoprecipitation, or proximity TurboID) do not always define direct interactions of the analyzed proteins. We specify this point in the new version of the manuscript to make it more clear to the readers.

We speculate that, even though localization by IF implies that LUZP1 is proximal and CCP110/CEP97 is distal on the centrioles, the centrosomal “environment” is in flux and dynamic. These proteins may encounter each other as steady-state patterns are established. Also, the proteins could be transported to the centrosome in complexes, which then disassemble and resolve into the patterns that we see by IF. LUZP1 and CCP110, as well as many other centrosomal proteins, are coiled-coil proteins that might be forming higher-order assemblies, so the interaction could be through bridging factors.

The model presented at the end of the paper focuses on LUZP1, SALL1, actin and ciliogenesis. How do CP110 and Cep97 fit to this model based on their data?

The destabilization of CCP110 at the mother centriole in absence of LUZP1 might be caused by the dysregulation of specific E3 ubiquitin ligases and deubiquitinases. Regulation of CCP110/CEP97 via the ubiquitin-proteasome system has been shown before (examples: Li et al., 2013; D'Angiolella et al., 2010; Wang et al. *eLife*. 2016. PMID: 27146717; Nagai et al. J Cell Sci. 2018. PMID: 30404837). In addition to the ones described to have a function at the centrosome, a higher number of E3s and DUBs have been recently identified at the centrosome by proteomics methods (https://cellmap.org/), increasing the landscape of potential centrosomal regulators. To understand how truncated SALL1 and/or LUZP1 modulates these E3s/DUBs would constitute a new project and cannot be done in the context of this manuscript. We integrated our data on CCP110 in our revised model in the new Figure 10D.

8) The authors switch between using RPE1 cells, U2OS cells, HEK293 cells, NIH3T3 cells and patient fibroblasts in different experiments. For the experiments related to LUZP1 phenotypic characterization, results for the same line should be included (additional cell lines can be kept as long as same one is carried along for all).

As with many published studies, different cell lines were used according to our particular objectives. For instance, HEK 293FT cells, where transient transfections are straightforward, were used for experiments that required decent protein levels (pulldowns). NIH3T3 ShhLight2 cells were used unmodified or for CRISPR-Cas9 LUZP1-knockouts, since they efficiently make primary cilia when starved and have a “built-in” *Shh*-responsive luciferase assay; U2OS cells do not form cilia, but have been used in many IF studies for centrosome/actin cytoskeleton. hTERT-RPE1 is a commonly used immortalized cell line that efficiently forms uniform primary cilia, mostly in a horizontal plane to allow easier length measurements. While they have nice primary cilia, RPE1 are poor or deficient in *Shh* signaling, which depends on many factors both upstream and downstream of the cilia. TBS patient-derived and control fibroblasts were used to demonstrate that the changes of LUZP1 are relevant in the context of the disease. While HEK 293FT cells can make primary cilia, they are poorly adherent and have few actin stress fibers, so they were rarely used for IF and quantifications of cellular structures. Aside from HEK 293FT, all other cells have much lower transfection efficiencies and sometimes lentiviral transductions were used. Considering the wide variety of techniques used in this manuscript, we had to use the most appropriate cell type for each technique.

At least for the LUZP1 localization by immunofluorescence, we believe that the same three localizations (actin filaments, centrosome, and midbody) have been observed in **all** cells used in the study throughout the course of our studies. We consider that using more than one cell type actually enriches the manuscript and makes our conclusions stronger.

[Editors’ note: what follows is the authors’ response to the second round of review.]

Essential revisions:The authors propose that LUZP1 is central to the mechanism linking cytoskeletal rearrangements and cilia assembly, which in turn impacts cell signaling and suggest that this underlies the etiology of TBS. While this speculation is certainly possible, the authors do not demonstrate a causative link of this mechanism to TBS. Additionally, the mechanism through which SALL1 and the proteasome regulate LUZP1 is not fully developed. In order for this manuscript to be acceptable for publication, the authors must modify the manuscript to very clearly indicate that they have not nailed the etiology of the disease. Furthermore, all suggestions about the therapeutic implications of their work should also be removed.

We are grateful for the feedback provided and now we present text revision, clarification, and new figures that we had already available to address the reviewers’ concerns. In brief, the main changes are the following:

– Mechanistic link between LUZP1 and Townes-Brocks Syndrome: we would like to clarify that it was not our intention to propose LUZP1 as the key regulator or the causative factor of TBS. Since truncated SALL1 acts dominantly and likely affects multiple factors, we believe that the reduction in LUZP1 levels might be one of the contributing factors in TBS etiology. As suggested by the reviewers, we clearly indicated this point in the new version of the manuscript. The link between LUZP1 and TBS is based in the following observations:

– Interaction of LUZP1 with truncated SALL1 by BioID, pulldown and immunoprecipitation (Figure 1 and Figure 1—figure supplement 2A).

– Truncated SALL1 can cause reduction of LUZP1 levels through ubiquitination and proteasomal degradation (WB: Figures 1A, 3D, Figure 9C-F), and in TBS cells (WB: Figure 9A-B; IF: Figures 3B, 4A, 10C).

– Cellular loss of LUZP1 promotes more and longer cilia, phenocopying TBS cells (Figure 6A, B, C).

– Loss of LUZP1 leads to a reduction of CCP110 levels in the mother centriole, phenocopying TBS cells (Figure 6D, E).

– Loss of LUZP1 leads to alterations in SHH pathway similar to TBS (Figure 7F-H).

– Importantly, cytoskeletal and cilia phenotypes in TBS cells are modified by exogenous LUZP1 (Figure 10).

– Mechanism through which SALL1 and the proteasome regulate LUZP1: many E3 ligases and deubiquitinases are known to function at the centrosome. To understand how truncated SALL1 exerts its effect on LUZP1 requires work beyond the scope of the current study. In the new version, we modified the Discussion explicitly stating, “Further experiments would be required to understand the precise mechanism by which truncated SALL1 can influence LUZP1 ubiquitination, but one possibility could be de novo complexes involving specific Ub E3 ligases or deubiquitinases which could influence LUZP1 stability”.

– Therapeutic implications of our findings: as proposed by the reviewers, we removed all the statements suggesting the potential therapeutic implications of our work.